# GRAPH TRANSFORMATION POLICY NETWORK FOR CHEMICAL REACTION PREDICTION

## ABSTRACT

We address a fundamental problem in chemistry known as chemical reaction product prediction. Our main insight is that the input reactant and reagent molecules can be jointly represented as a graph, and the process of generating product molecules from reactant molecules can be formulated as a sequence of graph transformations. To this end, we propose Graph Transformation Policy Network (GTPN) − a novel generic method that combines the strengths of graph neural networks and reinforcement learning to learn the reactions directly from data with minimal chemical knowledge. Compared to previous methods, GTPN has some appealing properties such as: end-to-end learning, and making no assumption about the length or the order of graph transformations. In order to guide model search through the complex discrete space of sets of bond changes effectively, we extend the standard policy gradient loss by adding useful constraints. Evaluation results show that GTPN improves the top-1 accuracy over the current state-of-the-art method by about 3% on the large *USPTO* dataset. Our model's performances and prediction errors are also analyzed carefully in the paper.

## 1 INTRODUCTION

Chemical reaction product prediction is a fundamental problem in organic chemistry. It paves the way for planning syntheses of new substances (Chen & Baldi, 2009). For decades, huge effort has been spent to solve this problem. However, most methods still depend on the handcrafted reaction rules (Chen & Baldi, 2009; Kayala & Baldi, 2011; Wei et al., 2016) or heuristically extracted reaction templates (Segler & Waller, 2017; Coley et al., 2017), thus are not well generalizable to unseen reactions.

A reaction can be regarded as a set (or unordered sequence) of graph transformations in which reactants represented as molecular graphs are transformed into products by modifying the bonds between some atom pairs (Jochum et al., 1980; Ugi et al., 1979). See Fig. 1 for an illustration. We call an atom pair $(u, v)$ that changes its connectivity during reaction and its new bond $b$ a *reaction triple* $(u, v, b)$. The reaction product prediction problem now becomes predicting a set of reaction triples given the input reactants and reagents. We argue that in order to solve this problem well, an intelligent system should have two key capabilities: (a) *Understanding the molecular graph structure* of the input reactants and reagents so that it can identify possible reactivity patterns (i.e., atom pairs with changing connectivity). (b) *Knowing how to choose from these reactivity patterns* a correct set of reaction triples to generate the desired products.

Recent state-of-the-art methods (Jin et al., 2017; Bradshaw et al., 2018) have built the first capability by leveraging graph neural networks (Duvenaud et al., 2015; Hamilton et al., 2017; Pham et al., 2017; Gilmer et al., 2017). However, these methods are either unaware of the valid sets of reaction triples (Jin et al., 2017) or limited to sequences of reaction triples with a predefined orders (Bradshaw et al., 2018). The main challenge is that the space of all possible configurations of reaction triples is extremely large and non-differentiable. Moreover, a small change in the predicted set of reaction triples can lead to very different reaction products and a little mistake can produce invalid prediction.

In this paper, we propose a novel method called Graph Transformation Policy Network (GTPN) that addresses the aforementioned challenges. Our model consists of three main components: a graph neural network (GNN), a node pair prediction network (NPPN) and a policy network (PN). Starting from the initial graph of reactant and reagent molecules, our model iteratively alternates between

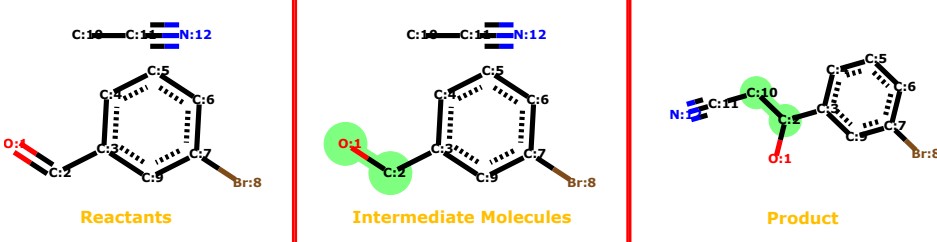

Figure 1: A sample reaction represented as a set of graph transformations from reactants (leftmost) to products (rightmost). Atoms are labeled with their type (Carbon, Oxygen,...) and their index (1, 2,...) in the molecular graph. The atom pairs that change connectivity and their new bonds (if existed) are highlighted in green. There are two bond changes in this case: 1) The double bond between O:1 and C:2 becomes single. 2) A new single bond between C:2 and C:10 is added.

modeling an input graph using GNN and predicting a reaction triple using NPPN and PN to generate a new intermediate graph as input for the next step until it decides to stop. The final generated graph is considered as the predicted products of the reaction. Importantly, GTPN does not assume any fixed number or any order of bond changes but learn these properties itself. One can view GTPN as a reinforcement learning (RL) agent that operates on a complex and non-differentiable space of sets of reaction triples. To guide our model towards learning a diverse yet robust-to-small-changes policy, we customize our loss function by adding some useful constraints to the standard policy gradient loss (Mnih et al., 2016).

To the best of our knowledge, GTPN is the most generic approach for the reaction product prediction problem so far in the sense that: i) It combines graph neural networks and reinforcement learning into a unified framework and trains everything end-to-end; ii) It does not use any handcrafted or heuristically extracted reaction rules/templates to predict the products. Instead, it automatically learns various types of reactions from the training data and can generalize to unseen reactions; iii) It can interpret how the products are formed via the sequence of reaction triples it generates.

We evaluate GTPN on two large public datasets named *USPTO-15k* and *USPTO*. Our method significantly outperforms all baselines in the top-1 accuracy, achieving new state-of-the-art results of 82.39% and 83.20% on *USPTO-15k* and *USPTO*, respectively. In addition, we also provide comprehensive analyses about the performance of GTPN and about different types of errors our model could make.

## 2 METHOD

### 2.1 CHEMICAL REACTION AS MARKOV DECISION PROCESS OF GRAPH TRANSFORMATIONS

A reaction occurs when reactant molecules interact with each other in the presence (or absence) of reagent molecules to form new product molecules by breaking or adding some of their bonds. Our main insight is that reaction product prediction can be formulated as predicting a sequence of such bond changes given the reactant and reagent molecules as input. A bond change is characterized by the atom pair (*where* the change happens) and the new bond type (*what* is the change). We call this atom pair a *reaction atom pair* and call this atom pair with the new bond type a *reaction triple*.

More formally, we represent the entire system of input reactant and reagent molecules as a labeled graph $\mathcal{G} = (\mathcal{V}, \mathcal{E})$ with multiple connected components, each of which corresponds to a molecule. Nodes in $\mathcal{V}$ are atoms labeled with their atomic numbers and edges in $\mathcal{E}$ are bonds labeled with their bond types. Given $\mathcal{G}$ as input, we predict a sequence of reaction triples that transforms $\mathcal{G}$ into a graph of product molecules $\mathcal{G}'$.

As reactions vary in number of transformation steps, we represent the sequence of reaction triples as $(\xi, u, v, b)^0, (\xi, u, v, b)^1, ..., (\xi, u, v, b)^{T-1}$ or $(\xi, u, v, b)^{0:T}$ for short. Here $T$ is the maximum number of steps, $(u, v)$ is a pair of nodes, $b$ is the new edge type of $(u, v)$, and $\xi$ is a binary signal that indicates the end of the sequence. If the sequence ends at $T_{\text{end}} < T$, $\xi^0, ...\xi^{T_{\text{end}}-1}$ will be 1 and $\xi^{T_{\text{end}}}, ..., \xi^{T-1}$ will be 0. At every step $\tau$, if $\xi^\tau = 1$, we apply the predicted edge change $(u, v, b)^\tau$ on

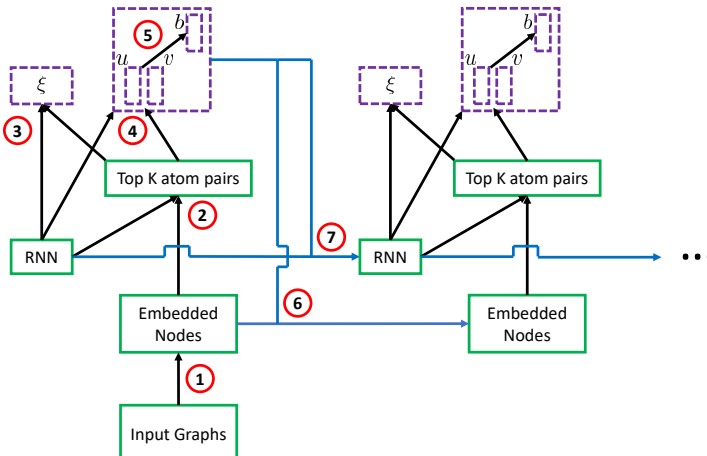

Figure 2: Workflow of a Graph Transformation Policy Network (GTPN). At every step of the forward pass, our model performs 7 major functions: 1) Computing the atom representation vectors, 2) Computing the most possible $K$ reaction atom pairs, 3) Predicting the continuation signal $\xi$, 4) Predicting the reaction atom pair $(u, v)$, 5) Predicting a new bond $b$ of this atom pair, 6) Updating the atom representation vectors, and 7) Updating the recurrent state.

the current graph $\mathcal{G}^\tau$ to create a new intermediate graph $\mathcal{G}^{\tau+1}$ as input for the next step $\tau + 1$. This iterative process of graph transformation can be formulated as a Markov Decision Process (MDP) characterized by a tuple $(\mathcal{S}, \mathcal{A}, P, R, \gamma)$, in which $\mathcal{S}$ is a set of states, $\mathcal{A}$ is a set of actions, $P$ is a state transition function, $R$ is a reward function, and $\gamma$ is a discount factor. Since the process is finite and contains no loop, we set the discount factor $\gamma$ to be 1. The rest of the MDP tuple are defined as follows:

- **State**: A state $s^\tau \in \mathcal{S}$ is an intermediate graph $\mathcal{G}^\tau$ generated at step $\tau$ ($0 \leq \tau < T$). When $\tau = 0$, we denote $s^0 = \mathcal{G}^0 = \mathcal{G}$.
- **Action**: An action $a^\tau \in \mathcal{A}$ performed at step $\tau$ is the tuple $(\xi, u, v, b)^\tau$. The action is composed of three consecutive sub-actions: $\xi^\tau$, $(u, v)^\tau$, and $b^\tau$. If $\xi^\tau = 0$, our model will ignore the next sub-actions $(u, v)^\tau$ and $b^\tau$, and all the future actions $(\xi, u, v, b)^{\tau+1:T}$. Note that setting $\xi^\tau$ to be the first sub-action is useful in case a reaction does not happen, i.e., $\xi^0 = 0$
- **State Transition**: If $\xi^\tau = 1$, the current graph $\mathcal{G}^\tau$ is modified based on the reaction triple $(u, v, b)^\tau$ to generate a new intermediate graph $\mathcal{G}^{\tau+1}$. We do not incorporate chemical rules such as valency check during state transition because the current bond change may result in invalid intermediate molecules $\mathcal{G}^\tau$, but later, other bond changes may compensate it to create the valid final products $\mathcal{G}^{T_{\text{end}}}$.
- **Reward**: We use both immediate rewards and delayed rewards to encourage our model to learn the optimal policy faster. At every step $\tau$, if the model predicts $\xi^\tau$, $(u, v)^\tau$ or $b^\tau$ correctly, it will receive a positive reward for each correct sub-action. Otherwise, a negative reward is given. After the prediction process has terminated, if the generated products are exactly the same as the groundtruth products, we give the model a positive reward, otherwise a negative reward. The concrete reward values are provided in Appendix A.3.

## 2.2 GRAPH TRANSFORMATION POLICY NETWORK

In this section, we describe the architecture of our model − a Graph Transformation Policy Network (GTPN). GTPN has three main components namely a Graph Neural Network (GNN), a Node Pair Prediciton Network (NPPN), and a Policy Network (PN). Each component is responsible for one or several key functions shown in Fig. 2: GNN performs functions 1 and 6; NPPN performs function 2; and PN performs functions 3, 4 and 5. Apart from these components, GTPN also has a Recurrent Neural Network (RNN) to keep track of the past transformations. The hidden state $h$ of this RNN is used by NPPN and PN to make accurate prediction.

### 2.2.1 Graph Neural Network

To model the intermediate graph $\mathcal{G}^\tau$ at step $\tau$, we compute the *node state vector* $\boldsymbol{x}_i^\tau$ of every node $i$ in $\mathcal{G}^\tau$ by using a variant of the Message Passing Neural Networks (Gilmer et al., 2017):

$$\boldsymbol{x}_i^\tau \;\;=\;\; \text{MessagePassing}^m\left(\boldsymbol{x}_i^{\tau-1}, \boldsymbol{v}_i, \mathcal{N}^\tau(i)\right) \tag{1}$$

where $m$ is the number of message passing steps; $\boldsymbol{v}_i$ is the feature vector of node $i$; $\mathcal{N}^\tau(i)$ is the set of all neighbor nodes of node $i$; and $\boldsymbol{x}_i^{\tau-1}$ is the state vector of node $i$ at the previous step. When $\tau = 0$, $\boldsymbol{x}_i^{\tau-1}$ is initialized from $\boldsymbol{v}_i$ using a neural network. Details about the MessagePassing(.) function are provided in Appendix A.1.

### 2.2.2 Node Pair Prediction Network

In order to predict how likely an atom pair $(i, j)$ of the intermediate graph $\mathcal{G}^\tau$ will change its bond, we assign $(i, j)$ with a score $s_{ij}^\tau \in \mathbb{R}$. If $s_{ij}^\tau$ is high, $(i, j)$ is more probably a reaction atom pair, otherwise, less probably. Similar to (Jin et al., 2017), we use two different networks called *"local"* network and *"global"* network for this task. In case of the *"local"* network, $s_{ij}^\tau$ is computed as:

$$\boldsymbol{z}_{ij}^\tau \;\;=\;\; \sigma\left(W_1\left[\boldsymbol{h}^{\tau-1}, (\boldsymbol{x}_i^\tau + \boldsymbol{x}_j^\tau), \boldsymbol{e}_{ij}\right] + b_1\right) \tag{2}$$
$$s_{ij}^\tau \;\;=\;\; f^{\text{atom pair}}\left(\boldsymbol{z}_{ij}^\tau\right) \tag{3}$$

where $f^{\text{atom pair}}$ is a neural network; $\sigma$ is a nonlinear activation function (e.g., ReLU); [.] denotes vector concatenation; $W_1$ and $b_1$ are parameters; $\boldsymbol{h}^{\tau-1}$ is the hidden state of the RNN at the previous step; and $\boldsymbol{e}_{ij}$ is the representation vector of the bond between $(i, j)$. If there is no bond between $(i, j)$ we assume that its bond type is *"NULL"*. We consider $\boldsymbol{z}_{ij}$ as the representation vector for the atom pair $(i, j)$.

The *"global"* network leverages self-attention (Vaswani et al., 2017; Wang et al., 2018) to detect compatibility between atom $i$ and all other atoms before computing the scores:

$$\boldsymbol{r}_{ij}^\tau \;\;=\;\; \sigma\left(V_1\left[(\boldsymbol{x}_i^\tau + \boldsymbol{x}_j^\tau), \boldsymbol{e}_{ij}\right] + c_1\right)$$
$$a_{ij}^\tau \;\;=\;\; \text{softmax}\left(V_2\boldsymbol{r}_{ij}^\tau + c_2\right)$$
$$\boldsymbol{c}_i^\tau \;\;=\;\; \sum_{j \in \mathcal{V}} a_{ij}\boldsymbol{x}_j^\tau$$
$$\boldsymbol{z}_{ij}^\tau \;\;=\;\; \sigma\left(W_1\left[\boldsymbol{h}^{\tau-1}, (\boldsymbol{x}_i^\tau + \boldsymbol{x}_j^\tau), (\boldsymbol{c}_i^\tau + \boldsymbol{c}_j^\tau), \boldsymbol{e}_{ij}\right] + b_1\right) \tag{4}$$
$$s_{ij}^\tau \;\;=\;\; f^{\text{atom pair}}\left(\boldsymbol{z}_{ij}^\tau\right) \tag{5}$$

where $a_{ij}$ is the attention score from node $i$ to every other node $j$; $\boldsymbol{c}_i$ is the context vector of atom $i$ that summarizes the information from all other atoms.

During experiments, we tried both options mentioned above and saw that the "global" network clearly outperforms the *"local"* network so we set the *"global"* network as a default module in our model. In addition, since reagents never change their form during a reaction, we explicitly exclude all atom pairs that have either atoms belong to the reagents. This leads to better results than not using reagent information. Detailed analyses are provided in Appendix A.5.

**Top-$K$ atom pairs**   Because the number of atom pairs that actually participate in a reaction is very small (usually smaller than 10) compared to the total number of atom pairs of the input molecules (usually hundreds or thousands), it is much more efficient to identify reaction triples from a small subset of highly probable reaction atom pairs. For that reason, we extract $K$ ($K \ll |\mathcal{V}|^2$) atom pairs with the highest scores. Later, we will predict reaction triples taken from these $K$ atom pairs only. We denote the set of top-$K$ atom pairs, their corresponding scores, and representation vectors as $\left\{(u_k, v_k)|k = \overline{1, K}\right\}$, $\left\{s_{u_k v_k}|k = \overline{1, K}\right\}$ and $Z_K = \left\{\boldsymbol{z}_{u_k v_k}|k = \overline{1, K}\right\}$, respectively.

### 2.2.3 Policy Network

**Predicting continuation signal**   To account for varying number of transformation steps, PN generates a continuation signal $\xi^\tau \in \{0, 1\}$ to indicate whether prediction should continue or terminate.

$\xi^\tau$ is drawn from a Bernoulli distribution:

$$p\left(\xi^\tau = 1\right) \quad = \quad \text{sigmoid}\left(f^{\text{signal}}\left(\left[\boldsymbol{h}^{\tau-1}, g\left(Z_K^\tau\right)\right]\right)\right) \tag{6}$$

where $\boldsymbol{h}^{\tau-1}$ is the previous RNN state; $Z_K^\tau$ is the set of representation vectors of the top $K$ atom pairs at the current step; $f^{\text{signal}}$ is a neural network; $g$ is a function that maps an unordered set of inputs to an output vector. For simplicity, we use a mean function:

$$\boldsymbol{z}_K^{\tau-1} = g\left(Z_K^\tau\right) = \frac{1}{K}\sum_{k=1}^{K} W\boldsymbol{z}_{u_k v_k}^{\tau-1}$$

**Predicting atom pair** At the next sub-step, PN predicts which atom pair changes its bond during the reaction by sampling from the top-$K$ atom pairs with probability:

$$p\left((u_k, v_k)^\tau\right) = \text{softmax}_K\left(s_{u_k v_k}^\tau\right) \tag{7}$$

where $s_{u_k v_k}^\tau$ is the score of the atom pair $(u_k, v_k)^\tau$ computed in Eq. (5). After predicting the atom pair $(u, v)^\tau$, we will mask it to ensure that it could not be in the top $K$ again at future steps.

**Predicting bond type** Given an atom pair $(u, v)^\tau$ sampled from the previous sub-step, we predict a new bond type $b^\tau$ between $u$ and $v$ to get a complete reaction triple $(u, v, b)^\tau$ using the probability:

$$p\left(b^\tau | (u, v)^\tau\right) = \text{softmax}_B\left(f^{\text{bond}}\left(\left[\boldsymbol{h}^{\tau-1}, \boldsymbol{z}_{uv}^\tau, (\boldsymbol{e}_b - \boldsymbol{e}_{b^{\text{old}}})\right]\right)\right) \tag{8}$$

where $B$ is the total number of bond types; $\boldsymbol{z}_{uv}^\tau$ is the representation vector of $(u, v)^\tau$ computed in Eq. (4); $b^{\text{old}}$ is the old bond of $(u, v)$; $\boldsymbol{e}_{b^{\text{old}}}$ and $\boldsymbol{e}_b$ are the embedding vectors corresponding to the bond type $b^{\text{old}}$ and $b$, respectively; and $f^{\text{bond}}$ is a neural network.

## 2.3 UPDATING STATES

After predicting a complete reaction triple $(u, v, b)^\tau$, our model updates: i) the new recurrent hidden state $\boldsymbol{h}^\tau$, and ii) the new node representation vectors $\boldsymbol{x}_i^{\tau+1}$ of the new intermediate graph $\mathcal{G}^{\tau+1}$ for $i \in \mathcal{V}$. These updates are presented in Appendix A.2.

## 2.4 TRAINING

Loss function plays a central role in achieving fast training and high performance. We design the following loss:

$$\mathcal{L} = \lambda_1 \mathcal{L}^{\text{A2C}} + \lambda_2 \mathcal{L}^{\text{value}} + \lambda_3 \mathcal{L}^{\text{atom pair}} + \lambda_4 \mathcal{L}^{\text{over length}} + \lambda_5 \mathcal{L}^{\text{in top } K}$$

where $\mathcal{L}^{\text{A2C}}$ is the Advantage Actor-Critic (A2C) loss (Mnih et al., 2016) to account for the correct sequence of reaction triples; $\mathcal{L}^{\text{value}}$ is the loss for estimating the value function used in A2C; $\mathcal{L}^{\text{atom pair}}$ accounts for binary change in the bond of an atom pair; $\mathcal{L}^{\text{over length}}$ penalizes long predicted sequences; and $\mathcal{L}^{\text{in top } K}$ is the rank loss to force a ground-truth reaction atom pair to appear in the top-$K$; and $\lambda_1, ..., \lambda_5 > 0$ are tunable coefficients. The component losses are explained in the following.

### 2.4.1 REACTION TRIPLE LOSS

The loss follows a policy gradient method known as Advantage Actor-Critic (A2C):

$$\begin{aligned} \mathcal{L}^{\text{A2C}} \quad = \quad &-\sum_{\tau=0}^{T_{\text{end}}-1}\left(A_{\text{signal}}^\tau \log p\left(\xi^\tau\right) + A_{\text{atom pair}}^\tau \log p\left((u, v)^\tau\right) + A_{\text{bond}}^\tau \log p\left(b^\tau\right)\right) \\ &-A_{\text{signal}}^{T_{\text{end}}} \log \pi\left(\xi^{T_{\text{end}}}\right) \end{aligned} \tag{9}$$

where $T_{\text{end}}$ is the first step that $\xi = 0$; $A_{\text{signal}}$, $A_{\text{atom pair}}$ and $A_{\text{bond}}$ are called *advantages*. To compute these advantages, we use the unbiased estimations called Temporal Different errors, defined as:

$$A_{\text{signal}}^{\tau} = r_{\text{signal}}^{\tau} + \gamma V_{\phi}\left(Z_{K}^{\tau+1}\right) - V_{\phi}\left(Z_{K}^{\tau}\right) \tag{10}$$

$$A_{\text{atom pair}}^{\tau} = r_{\text{atom pair}}^{\tau} + \gamma V_{\phi}\left(Z_{K}^{\tau+1}\right) - V_{\phi}\left(Z_{K}^{\tau}\right) \tag{11}$$

$$A_{\text{bond}}^{\tau} = r_{\text{bond}}^{\tau} + \gamma V_{\phi}\left(Z_{K}^{\tau+1}\right) - V_{\phi}\left(Z_{K}^{\tau}\right) \tag{12}$$

where $r_{\text{signal}}^{\tau}$, $r_{\text{atom pair}}^{\tau}$, $r_{\text{bond}}^{\tau}$ are immediate rewards at step $\tau$; at the final step $\tau = T_{\text{end}}$, the model receives additional delayed rewards; $\gamma$ is the discount factor; and $V_{\phi}$ is the parametric value function. We train $V_{\phi}$ using the following mean square error loss:

$$\mathcal{L}^{\text{value}} = \sum_{\tau=0}^{T_{\text{end}}} \|V_{\phi}\left(Z_{K}^{\tau}\right) - R^{\tau}\|^{2} \tag{13}$$

where $R^{\tau}$ is the return at step $\tau$.

**Episode termination during training** Although the loss defined in Eq. (9) is correct, it is not good to use in practice because: i) If our model selects a wrong sub-action at any sub-step of the step $T_{\text{wrong}}$ ($T_{\text{wrong}} < T_{\text{end}}$), the whole predicted sequence will be incorrect regardless of what will be predicted from $T_{\text{wrong}} + 1$ to $T_{\text{end}}$. Therefore, computing the loss for actions from $T_{\text{wrong}} + 1$ to $T_{\text{end}}$ is redundant. ii) More importantly, the incorrect updates of the graph structure at subsequent steps from $T_{\text{wrong}} + 1$ to $T_{\text{end}}$ will lead to cumulative prediction errors which make the training of our model much more difficult.

To resolve this issue, during training, we use a binary vector $\boldsymbol{\zeta} \in \{0,1\}^{3T}$ to keep track of the first wrong sub-action: $\zeta^{t} = \begin{cases} 1 & \text{if } t \leq t_{\text{first wrong}} \\ 0 & \text{if } t > t_{\text{first wrong}} \end{cases}$ where $t_{\text{first wrong}}$ denotes the sub-step at which our model chooses a wrong sub-action the first time. The actor-critic loss in Eq. (9) now becomes:

$$\mathcal{L}^{\text{A2C}} = -\sum_{\tau=0}^{T}\left(\zeta^{\tau} A_{\text{signal}}^{\tau} \log p\left(\xi^{\tau}\right) + \zeta^{(\tau+1)} A_{\text{atom pair}}^{\tau} \log p\left((u,v)^{\tau}\right) + \zeta^{(\tau+2)} A_{\text{bond}}^{\tau} \log p\left(b^{\tau}\right)\right)$$
$$\tag{14}$$

where $T$ is the maximum number of steps. Similarly, we change the value loss into:

$$\mathcal{L}^{\text{value}} = \sum_{\tau=0}^{T} \zeta^{\tau} \|V_{\phi}\left(Z_{K}^{\tau}\right) - R^{\tau}\|^{2}$$

### 2.4.2 REACTION ATOM PAIR LOSS

To train our model to assign higher scores to reaction atom pairs and lower to non-reaction atom pairs, we use the following cross-entropy loss function:

$$\mathcal{L}^{\text{atom pair}} = -\sum_{\tau=0}^{T_{\text{first wrong}}} \sum_{i\in\mathcal{V}} \sum_{j\in\mathcal{V}, j\neq i} \eta_{ij\tau}\left(y_{ij}\log p_{ij} + (1-y_{ij})\log(1-p_{ij})\right) \tag{15}$$

where $T_{\text{first wrong}} = \left\lfloor \frac{t_{\text{first wrong}}}{3} \right\rfloor$; $\eta_{ijt} \in \{0,1\}$ is a mask of the atom pair $(i,j)$ at step $\tau$; $y_{ij} \in \{0,1\}$ is the label indicating whether the atom pair $(i,j)$ is a reaction atom pair or not; $p_{ij} = \text{sigmoid}(s_{ij})$ (see Eq. (5)).

### 2.4.3 CONSTRAINT ON THE SEQUENCE LENGTH

One major difficulty of the chemical reaction prediction problem is to know exactly when to stop prediction so we can make accurate inference. By forcing the model to stop immediately when making wrong prediction, we can prevent cumulative error and significantly reduce variance during training. But it also comes with a cost: The model cannot learn (because it does not have to learn) when to stop. This phenomenon can be visualized easily as the model predicts 1 for the signal at

| Dataset | | #reactions | #changes | #molecules | #atoms | #bonds |
|---------|-------|-----------|----------|------------|--------|--------|
| *USPTO-15k* | train | 10,500 | 1 \| 11 \| 2.3 | 1 \| 20 \| 3.6 | 4 \| 100 \| 34.9 | 3 \| 110 \| 34.7 |
| | valid | 1,500 | 1 \| 11 \| 2.3 | 1 \| 20 \| 3.6 | 7 \| 94 \| 34.5 | 5 \| 99 \| 34.2 |
| | test | 3,000 | 1 \| 11 \| 2.3 | 1 \| 16 \| 3.6 | 7 \| 98 \| 34.9 | 5 \| 102 \| 34.7 |
| *USPTO* | train | 409,035 | 1 \| 6 \| 2.2 | 2 \| 29 \| 4.8 | 9 \| 150 \| 39.7 | 6 \| 165 \| 38.6 |
| | valid | 30,000 | 1 \| 6 \| 2.2 | 2 \| 25 \| 4.8 | 9 \| 150 \| 39.6 | 7 \| 158 \| 38.5 |
| | test | 40,000 | 1 \| 6 \| 2.2 | 2 \| 22 \| 4.8 | 9 \| 150 \| 39.8 | 7 \| 162 \| 38.7 |

Table 1: Statistics of *USPTO-15k* and *USPTO* datasets. *"changes"* means bond changes, *"molecules"* means reactants and reagents in a reaction; *"atoms"* and *"bonds"* are defined for a molecule. Apart from *"#reactions"*, other columns are presented in the format "min | max | mean".

every step $\tau$ during inference. In order to make the model aware of the correct sequence length during training, we define a loss that punishes the model if it produces a longer sequence than the ground truth sequence:

$$\mathcal{L}^{\text{over length}} = - \sum_{T_{\text{end}}^{\text{gt}} \leq \tau < T_{\text{end}}} \log p \left( \xi^\tau = 0 \right) \qquad (16)$$

where $T_{\text{end}}^{\text{gt}}$ is the end step of the ground-truth sequence. Note that the loss in Eq. (16) is not applied when $T_{\text{end}} \leq T_{\text{end}}^{\text{gt}}$. The reason is that forcing $\xi^\tau = 1$ with $T_{\text{end}} \leq \tau < T_{\text{end}}^{\text{gt}}$ is not theoretically correct because all the signals after $T_{\text{end}}$ are assumed to be 0. The incentive to force $T_{\text{end}}$ close to $T_{\text{end}}^{\text{gt}}$ when it is smaller than $T_{\text{end}}^{\text{gt}}$ has already been included in the advantages in Eq. (14).

### 2.4.4 CONSTRAINT ON THE TOP-$K$ ATOM PAIRS

Ideally, the loss from Eq. (15) pushes a reaction atom pair $(\tilde{u}, \tilde{v})^\tau$ into the top-$K$ atom pairs at each step $\tau < T_{\text{end}}^{\text{gt}}$. However, this is not guaranteed, especially when $\tau$ comes close to $T_{\text{end}}^{\text{gt}}$. To encourage the ground-truth reaction atom pair $(\tilde{u}, \tilde{v})^\tau$ with the highest score to appear in the top $K$, we introduce an additional rank-based loss:

$$\mathcal{L}^{\text{in top } K} = - \sum_{\tau=0}^{T_{\text{first wrong}}} \log p \left( (\tilde{u}, \tilde{v})^\tau \text{ in top } K \right)$$

where $p \left( (\tilde{u}, \tilde{v})^\tau \text{ in top } K \right)$ is computed as:

$$p \left( (\tilde{u}, \tilde{v})^\tau \text{ in top } K \right) = \frac{\exp \left( s_{\tilde{u}\tilde{v}}^\tau \right)}{\exp \left( s_{\tilde{u}\tilde{v}}^\tau \right) + \sum_{k=1}^{K} \exp \left( s_{u_k v_k}^\tau \right)} \qquad (17)$$

## 3 EXPERIMENTS

### 3.1 DATASET

We evaluate our model on two standard datasets *USPTO-15k* (15K reactions) and *USPTO* (480K reactions) which have been used in previous works (Jin et al., 2017; Schwaller et al., 2018; Bradshaw et al., 2018). Details about these datasets are given in Table 1. The USPTO dataset contains reactant, reagent and product molecules represented as SMILES strings. Using RDKit[1], we convert the SMILES strings into molecule objects and store them as graphs. For each reaction, every atom in the reactant and reagent molecules is identified with a unique "atom map number". This identity is the same in the products. Using this knowledge, we compare every atom pair in the input molecules with the correspondent in the product molecules to obtain a ground-truth set of reaction triples for training. In *USPTO-15k*, the ground-truth sets of reaction triples was precomputed by (Jin et al., 2017).

---

[1]https://www.rdkit.org/

| Model | USPTO-15k | | | USPTO | | |
|---|---|---|---|---|---|---|
| | C@6 | C@8 | C@10 | C@6 | C@8 | C@10 |
| WLN⋆ (Jin et al., 2017) | 81.6 | 86.1 | 89.1 | 89.8 | 92.0 | 93.3 |
| WLN (Jin et al., 2017) | 88.45 | 91.65 | 93.34 | 90.97 | 93.98 | 95.26 |
| CLN (Pham et al., 2017) | 88.68 | 91.63 | 93.07 | 90.72 | 93.57 | 94.80 |
| Our GNN | **88.92** | **92.00** | **93.57** | **91.24** | **94.17** | **95.33** |

Table 2: Results for reaction atom pair prediction. $C@k$ is coverage at $k$. Best results are highlighted in bold. WLN⋆ is the original model from (Jin et al., 2017) while WLN is our re-implemented version. Except for WLN⋆, other models explicitly use reagent information.

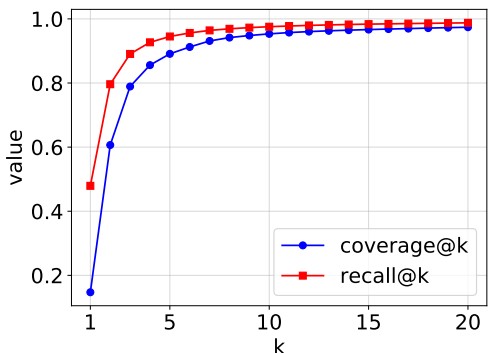

Figure 3: *Coverage@k* and *Recall@k* with respect to $k$ for the *USPTO* dataset.

## 3.2 REACTION ATOM PAIR PREDICTION

In this section, we test our model's ability to identify reaction atom pairs by formulating it as a ranking problem with the scores computed in Eq. (5). Similar to (Jin et al., 2017), we use *Coverage@k* as the evaluation metric, which is the proportion of reactions that have *all* groundtruth reaction atom pairs appear in the top $k$ predicted atom pairs.

We compare our proposed graph neural network (GNN) with Weisfeiler-Lehman Network (WLN) (Jin et al., 2017) and Column Network (CLN) (Pham et al., 2017). Since our GNN explicitly uses reagent information to compute the scores of atom pairs, we modify the implementation of WLN and CLN accordingly for fair comparison. From Table 2, we observe that our GNN clearly outperforms WLN and CLN in all cases. We attribute this improvement to the use of a separate node state vector $\boldsymbol{x}_i^t$ (different from the node feature vector $\boldsymbol{v}_i$) for updating the structural information of a node (see Eq. (21)). The other two models, on the other hand, only use a single vector to store both the node features and structure, hence, some information may be lost. In addition, using explicit reagent information boosts the prediction accuracy, which improves the WLN by 1-7% depending on the metrics. The presence of reagent information reduces the number of atom pairs to be searched on and contributes to the likelihood of reaction atom pairs. Further results are presented in Appendix A.5.

## 3.3 TOP-$K$ ATOM PAIR EXTRACTION

The performance of our model depends on the number of selected top atom pairs $K$. The value of $K$ presents a trade-off between coverage and efficiency. In addition to the metric *Coverage@k* in Sec. 3.2, we use *Recall@k* which is the proportion of correct atom pairs that appear in top $k$ to find the good $K$. Fig. 3 shows *Coverage@k* and *Recall@k* for the *USPTO* dataset with respect to $k$. We see that both curves increase rapidly when $k < 10$ and stablize when $k > 10$. We also ran experiments with $k = 10, 15, 20$ and observed that their prediction results are quite similar. Hence, in what follows we select $K = 10$ for efficiency.

| Model | USPTO-15k | | | USPTO | | |
|---|---|---|---|---|---|---|
| | P@1 | P@3 | P@5 | P@1 | P@3 | P@5 |
| WLDN (Jin et al., 2017) | *76.7* | *85.6* | ***86.8*** | *79.6* | ***87.7*** | ***89.2*** |
| Seq2Seq (Schwaller et al., 2018) | - | - | - | 80.3* | 86.2* | 87.5* |
| GTPN | 72.31 | - | - | 71.26 | - | - |
| GTPN$^\diamond$ | 74.56 | 82.62 | 84.23 | 73.25 | 80.56 | 83.53 |
| GTPN$^{\diamond\clubsuit}$ | 74.56 | 83.19 | 84.97 | 73.25 | 84.31 | 85.76 |
| GTPN$^{\diamond\spadesuit}$ | **82.39** | 85.60 | 86.68 | **83.20** | 84.97 | 85.90 |
| GTPN$^{\diamond\spadesuit\clubsuit}$ | **82.39** | **85.73** | **86.78** | **83.20** | 86.03 | 86.48 |

Table 3: Results for reaction prediction. *P@k* is precision at $k$. State-of-the-art results from (Jin et al., 2017) are written in italic. Results from (Schwaller et al., 2018) are marked with $\star$ and they are computed on a slightly different version of USPTO that contains only single-product reactions. Best results are highlighted in bold. $\diamond$: With beam search (beam width = 20), $\spadesuit$: Invalid product removal, $\clubsuit$: Duplicated product removal.

## 3.4 REACTION PRODUCT PREDICTION

This experiment validates GTPN on full reaction product prediction against the recent state-of-the-art methods (Jin et al., 2017; Schwaller et al., 2018) using the accuracy metric. The recent method ELECTRO (Bradshaw et al., 2018) is not compatible here because it was only evaluated on a subset of *USPTO* limited to linear chain topology. Comparison against ELECTRO is reported separately in Appendix A.6. Table 3 shows the prediction results. We produce multiple reaction product candidates by using beam search decoding with beam width $N = 20$. Details about beam search and its behaviors are presented in Appendix A.4.

In brief, we compute the normalized-over-length log probabilities of $N$ predicted sequences of reaction triples and sort these values in descending order to get a rank list of $N$ possible reaction outcomes. Given a predicted sequence of reaction triples $(u, v, b)^{0:T}$, we can generate reaction products from input reactants simply by replacing the old bond of $(u, v)^\tau$ with $b^\tau$. However, these products are not guaranteed to be valid (e.g., maximum valence constraint violation or aromatic molecules cannot be kekulized) so we post-process the outputs by removing all invalid products. The removal increases the top-1 accuracy by about 8% and 10% on *USPTO-15k* and *USPTO*, respectively. Due to the permutation invariance of the predicted sequence of reaction triples, some product candidates are duplicate and will also be removed. This does not lead to any change in *P@1* but slightly improves *P@3* and *P@5* by about 0.5-1% on the two datasets.

Overall, GTPN with beam search and post-processing outperforms both WLDN (Jin et al., 2017) and Seq2Seq (Schwaller et al., 2018) in the top-1 accuracy. For the top-3 and top-5, our model's performance is comparable to WLDN's on *USPTO-15k* and is worse than WLDN's on *USPTO*. It is not surprising since our model is trained to accurately predict the top-1 outcomes instead of ranking the candidates directly like WLDN. It is important to emphasize that we did not tune the model hyper-parameters when training on *USPTO* but reused the optimal settings from *USPTO-15k* (which is 25 times smaller than *USPTO*) so the results may not be optimal (see Appendix A.3 for more training detail).

## 4 RELATED WORK

### 4.1 LEARNING TO PREDICT CHEMICAL REACTION

In chemical reaction prediction, machine learning has replaced rule-based methods (Chen & Baldi, 2009) for better generalizability and scalability. Existing machine learning-based techiques are either template-free (Kayala & Baldi, 2011; Jin et al., 2017; Fooshee et al., 2018) and template-based (Wei et al., 2016; Segler & Waller, 2017; Coley et al., 2017). Both groups share the same mechanism: running multiple stages with the aid of reaction templates or rules. For example, in (Wei et al., 2016) the authors proposed a two-stage model that first classifies reactions into different types based on the neural fingerprint vectors (Duvenaud et al., 2015) of reactant and reagent molecules. Then, it applies

pre-designed SMARTS transformation on the reactants with respect to the most suitable predicted reaction type to generate the reaction products.

The work of (Jin et al., 2017) treats a reaction as a set of bond changes so in the first step, they predict which atom pairs are likely to be reactive using a variant of graph neural networks called Weisfeiler-Lehman Networks (WLNs). In the next step, they do almost the same as (Coley et al., 2017) by modifying the bond type between the selected atom pairs (with chemical rules satisfied) to create product candidates and rank them (with reactant molecules as addition input) using another kind of WLNs called Weifeiler-Lehman Different Networks (WLDNs).

To the best of our knowledge, (Jin et al., 2017) is the first work that achieves remarkable results (with the Precision@1 is about 79.6%) on the large *USPTO* dataset containing more than 480 thousands reactions. Works of (Nam & Kim, 2016) and (Schwaller et al., 2018) avoid multi-stage prediction by building a seq2seq model that generates the (canonical) SMILES string of the single product from the concatenated SMILES strings of the reactants and reagents in an end-to-end manner. However, their methods cannot deal with sets of reactants/reagents/products properly as well as cannot provide concrete reaction mechanism for every reaction.

The most recent work on this topic is (Bradshaw et al., 2018) which solves the reaction prediction problem by predicting a sequence of bond changes given input reactants and reagents represented as graphs. To handle ordering, they only select reactions with predefined topology. Our method, by contrast, is order-free and can be applied to almost any kind of reactions.

## 4.2 GRAPH NEURAL NETWORKS FOR MODELING MOLECULES

In recent years, there has been a fast development of graph neural networks (GNNs) for modeling molecules. These models are proposed to solve different problems in chemistry including toxicity prediction (Duvenaud et al., 2015), drug activity classification (Shervashidze et al., 2011; Dai et al., 2016; Pham et al., 2018), protein interface prediction (Fout et al., 2017) and drug generation (Simonovsky & Komodakis, 2018; Jin et al., 2018). Most of them can be regarded as variants of message-passing graph neural networks (MPGNNs) (Gilmer et al., 2017).

## 4.3 REINFORCEMENT LEARNING FOR STRUCTURAL REASONING

Reinforcement learning (RL) has become a standard approach to many structural reasoning problems[2] because it allows agents to perform discrete actions. A typical example of using RL for structural reasoning is drug generation (Li et al., 2018; You et al., 2018). Both (Li et al., 2018) and (You et al., 2018) learn the same generative policy whose action set including: i) adding a new atom or a molecular scaffold to the intermediate graph, ii) connecting existing pair of atoms with bonds, and iii) terminating generation. However, (You et al., 2018) uses an adversarial loss to enforce global chemical constraints on the generated molecules as a whole instead of using the common reconstruction loss as in (Li et al., 2018). Other examples are path-based relational reasoning in knowledge graphs (Das et al., 2018) and learning combinatorial optimization over graphs (Khalil et al., 2017).

## 5 DISCUSSION

We have introduced a novel method named Graph Transformation Policy Network (GTPN) for predicting products of a chemical reaction. GTPN uses graph neural networks to represent input reactant and reagent molecules, and uses reinforcement learning to find an optimal sequence of bond changes that transforms the reactants into products. We train GTPN using the Advantage Actor-Critic (A2C) method with appropriate constraints to account for notable aspects of chemical reaction. Experiments on real datasets have demonstrated the competitiveness of our model.

Although the GTPN was proposed to solve the chemical reaction problem, it is indeed generic to solve the graph transformation problem, which can be useful in reasoning about relations (e.g., see (Zambaldi et al., 2018)) and changes in relation. Open rooms include addressing dynamic graphs over time, extending toward full chemical planning and structural reasoning using RL.

---

[2]Structural reasoning is a problem of inferring or generating new structure (e.g. objects with relations)

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

# A  APPENDIX

## A.1  GRAPH NEURAL NETWORK

In this section, we describe our graph neural network (GNN) in detail. Since our GNN does not use the recurrent hidden state $h^\tau$, we exclude the time step $\tau$ from our notations for clarity. Instead, we use $t$ to denote a message passing step.

### GRAPH NOTATIONS

Input to our GNN is a graph $\mathcal{G} = (\mathcal{V}, \mathcal{E})$ in which each node $i \in \mathcal{V}$ is represented by a node feature vector $v_i$ and each edge $(i, j) \in \mathcal{E}$ is represented by an edge feature vector $e_{ij}$. For example of molecular graph, the node feature vector $v_i$ may include chemical information about the atom $i$ such as its type, charge and degree. Similarly, $e_{ij}$ captures the bond type between the two atoms $i$ and $j$. We denote by $\mathcal{N}(i)$ the set of all neighbor nodes of node $i$ together with their links to node $i$:

$$\mathcal{N}(i) \quad \equiv \quad \{(j, e_{ij}) \mid j \text{ is a neighbor node of } i\}$$

If we only care about the neighbor nodes of $i$ not their links, we use the notation $\mathcal{N}_n(i)$ defined as:

$$\mathcal{N}_n(i) \quad \equiv \quad \{j \mid j \text{ is a neighbor node of } i\}$$

In addition to $v_i$, node $i$ also has a state vector $x_i$ to store information about itself and the surrounding context. This state vector is updated recursively using the neural message passing method (Battaglia et al., 2016; Pham et al., 2017; Hamilton et al., 2017; Gilmer et al., 2017; Schlichtkrull et al., 2018). The initial state $x_i^0$ is the nonlinear mapping of $v_i$:

$$x_i^0 \quad = \quad \sigma\left(W v_i + b\right) \tag{18}$$

### COMPUTING NEIGHBOR MESSAGES

At the message passing step $t$, we compute the message $m_{ij}^t$ from every neighbor node $j \in \mathcal{N}_n(i)$ to node $i$ as:

$$
\begin{aligned}
m_{ij}^t \quad &= \quad f\left(x_i^t, x_j^t, e_{ij}\right) \\
&= \quad \sigma\left(W\left[x_i^t, x_j^t, e_{ij}\right] + b\right)
\end{aligned}
\tag{19}
$$

where $[\cdot]$ denotes concatenation; and $\sigma$ is a nonlinear function.

### AGGREGATING NEIGHBOR MESSAGES

Then, we aggregate all the messages sent to node $i$ into a single message vector by averaging:

$$m_i^t \quad = \quad \frac{1}{|\mathcal{N}_n(i)|} \sum_{j \in \mathcal{N}_n(i)} m_{ij}^t \tag{20}$$

where $|\mathcal{N}_n(i)|$ is the number of neighbor nodes of node $i$.

### UPDATING NODE STATE

Finally, we update the state of node $i$ as follows:

$$x_i^{t+1} \quad = \quad g\left(x_i^t, m_i^t, v_i\right) \tag{21}$$

where $g(.)$ is a Highway Network (Srivastava et al., 2015):

$$
\begin{aligned}
x_i^{t+1} \quad &= \quad \text{Highway}\left(x_i^t, m_i^t, v_i\right) \tag{22} \\
&= \quad \alpha * \tilde{x}_i^{t+1} + (1 - \alpha) * x_i^t \tag{23}
\end{aligned}
$$

where $\tilde{\boldsymbol{x}}_i^{t+1}$ is the nonlinear part which is computed as:
$$\bar{\boldsymbol{x}}_i^{t+1} = \sigma \left( W_1 \left[ \boldsymbol{x}_i^t, \boldsymbol{m}_i^t, \boldsymbol{v}_i^t \right] + b_1 \right)$$
and $\boldsymbol{\alpha}$ is the gate controlling the flow of information:

$$\boldsymbol{\alpha} = \text{sigmoid}(W_2 \left[ \boldsymbol{x}_i^t, \boldsymbol{m}_i^t, \boldsymbol{v}_i^t \right] + b_2)$$

By combining Eqs. (19,20,22) together, one step of message passing update for node $i$ can be written in a generic way as follows:

$$\boldsymbol{x}_i^{t+1} = \text{MessagePassing} \left( \boldsymbol{x}_i^t, \boldsymbol{v}_i, \mathcal{N}(i) \right) \qquad (24)$$

## A.2 UPDATING STATES

### UPDATING RNN STATE

We keep the old representation of the edge that have been modified in the hidden memory of the RNN as follows:

$$\boldsymbol{h}^\tau = \text{GRU} \left( \boldsymbol{h}^{\tau-1}, \boldsymbol{z}_{uv}^\tau \right) \qquad (25)$$

where GRU stands for Gated Recurrent Units (Cho et al., 2014); $\boldsymbol{z}_{uv}^\tau$ is the representation vector of the atom pair $(u, v)^\tau$ including its old bond (see Eq. 4). Eq. (25) allows the model to keep track of all the changes happening to the graph so far so it can make more accurate prediction later.

### UPDATING GRAPH STRUCTURE AND NODE STATES

After predicting a reaction triple $(u, v, b)^\tau$ at step $\tau$, we update the graph structure and node states based on the new bond change. First, to update the graph structure, we simply update the neighbor set of $u$ and $v$ with information from the other atom and the new bond type $b$ as follows:

$$\mathcal{N}^\tau(u) = \left( \mathcal{N}^{\tau-1}(u) \backslash \left( v, b^{\text{old}} \right) \right) \cup (v, b) \qquad (26)$$
$$\mathcal{N}^\tau(v) = \left( \mathcal{N}^{\tau-1}(v) \backslash \left( u, b^{\text{old}} \right) \right) \cup (u, b) \qquad (27)$$

Next, to update the node states, our model performs one step of message passing for $u$ and $v$ with their new neighbor sets:

$$\boldsymbol{x}_u^\tau = \text{MessagePassing} \left( \boldsymbol{x}_u^{\tau-1}, \boldsymbol{v}_u, \mathcal{N}^\tau(u) \right) \qquad (28)$$
$$\boldsymbol{x}_v^\tau = \text{MessagePassing} \left( \boldsymbol{x}_v^{\tau-1}, \boldsymbol{v}_v, \mathcal{N}^\tau(v) \right) \qquad (29)$$

where the MessagePassing(.) function is defined in Eq. (24). For other nodes in the graph to be aware of the new structures of $u$ and $v$, we need to perform several message passing steps for all nodes in the graph after Eqs. (28, 29). However, it is very costly to run for every prediction step $\tau$. Sometimes it is unnecessary since far-away bonds are less likely to be affected by the current bond change (unless the far-way bonds and the new bond are in an aromatic ring). Therefore, in our model, we limit the number of message passing updates for all nodes at step $\tau$ to be 1.

## A.3 MODEL CONFIGURATIONS

We optimize our model's hyper-parameters in two stages: First, we tune the hyper-parameters of the GNN and the NPPN for the reaction atom pair prediction task. Then, we fix the optimal settings of the first two components and optimize the hyper-parameters of the PN for the reaction product prediction task.

We provide details about the settings that give good results on the *USPTO-15k* dataset below. With these settings, we trained another model on the *USPTO* dataset from scratch. Because training on the large dataset such as the *USPTO* takes time, we did not tune hyper-parameters on the *USPTO*, eventhough it is possible to increase model sizes for better performance.

Unless explicitly stated, all neural networks in our model have 2 layers with the same number of hidden units, ReLU activation and residual connections (He et al., 2016).

| Atom attribute | Data type |
|---|---|
| Degree | numeric |
| Explicit valence | numeric |
| Explicit number of Hs | numeric |
| Charge | numeric |
| Part of a ring | boolean |

Table 4: Data types of atom attributes.

**Graph Neural Network (GNN)** There are 72 different types of atom depending on their atomic numbers and 5 different types of bond including NULL, SINGLE, DOUBLE, TRIPLE and ARO-MATIC. The size of embedding vectors for atom and bond are 51 and 21, respectively. Apart from atom type, each atom has 5 more attributes listed in Table 4. These attributes are normalized to the range of [0, 1] and are concatenated to the atom embedding vector to form a final atom feature vector of size 56. The state vector and the neighbor message vector for an atom both have the size of 99. The number of message passing steps is 6.

**Node Pair Prediction Network (NPPN)** This component consists of two parts. The first part computes the representation vector $z_{ij}$ of an atom pair $(i, j)$ using a neural network with hidden size of 71. The second part maps $z_{ij}$ to an unnormalized score $s_{ij}$ using the function $f^{\text{atom pair}}$ (see Eqs. (3,5)). This function is also a neural network with hidden size of 51.

**Policy Network (PN)** The recurrent network is a GRU (Cho et al., 2014) with 101 hidden units. The value function $V_\phi$ is a neural network with 99 hidden units. The two functions $f^{\text{signal}}$ for computing signal scores (see Eq. (6)) and $f^{\text{bond}}$ for computing scores over bond types (see Eq. (8)) are neural networks with 81 hidden units.

**Training** At each step, we set the reward to be 1.0 for correct prediction of signal/atom pair/bond type and -1.0 for incorrect prediction. After the prediction sequence is terminated (zero signal was emitted), we check whether the entire set of predicted reaction triples is correct or not. If it is correct, we give the model a reward value of 2.0, otherwise -2.0. From the rewards and estimated values for signal, atom pair and bond type, we define the Advantage Actor Critic loss (A2C) as in Eq. (14). The coefficients of components in the final loss $\mathcal{L}$ are set empirically as follows:

$$\mathcal{L} = \mathcal{L}^{\text{A2C}} + 0.5 \times \mathcal{L}^{\text{value}} + \mathcal{L}^{\text{atom pair}} + 0.2 \times \mathcal{L}^{\text{over length}} + 0.2 \times \mathcal{L}^{\text{in top } K}$$

We trained our model using Adam (Kingma & Ba, 2015) with the initial learning rate of 0.001 for both *USPTO-15k* and *USPTO*. For *USPTO-15k*, the learning rate will decrease by half if the *Precision@1* does not improve on the validation set after 1,000 steps until it reaches the minimum value of $5 \times 10^{-5}$. For *USPTO*, the decay rate is 0.8 after every 500 steps of no improvement until reaching the minimum learning rate is $2 \times 10^{-5}$. The maximum number of training iterations is $10^6$ and the batch size is 20.

A.4    DECODING WITH BEAM SEARCH

For decoding, our model generates a sequence of reaction triples (including the stop signal) $(\xi, u, v, b)$ by taking the best $(u, v)$ and $b$ at every step until it outputs a zero signal ($\xi = 0$). In other words, it computes the argmax of $p\left((\xi, u, v, b)^\tau \mid \mathcal{G}, (\xi, u, v, b)^{0:\tau-1}\right)$ at every step $\tau$. However, this algorithm is not robust for the sequence generation task because just a single error at a step may destroy the entire sequence. To overcome this issue, we employ beam search for decoding.

During beam search, we keep track of $N > 1$ best subsequences at every step $\tau$. $N$ is called beam width. Instead of modeling the conditional distribution of generating an output at the current step $\tau$, we model the joint distribution of the whole subsequence that has been generated from 0 to $\tau$:

$$\log p\left((\xi, u, v, b)^{0:\tau}|\mathcal{G}\right) = \log p\left((\xi, u, v, b)^\tau|\mathcal{G}, (\xi, u, v, b)^{0:\tau-1}\right) + \\ \log p\left((\xi, u, v, b)^{0:\tau-1}|\mathcal{G}\right) \tag{30}$$

| Beam width | Precision@k | | | | | | |
|---|---|---|---|---|---|---|---|
| | 1 | 2 | 3 | 5 | 10 | 15 | 20 |
| 1 | 74.49 | - | - | - | - | - | - |
| 2 | 72.21 | 80.65 | - | - | - | - | - |
| 5 | 72.21 | 79.54 | 82.29 | **84.27** | - | - | - |
| 10 | 72.15 | 79.54 | 82.19 | 83.93 | 86.01 | - | - |
| 15 | 72.15 | 79.54 | 82.16 | 83.93 | 86.11 | 86.98 | - |
| 20 | **74.56** | **80.72** | **82.62** | 84.23 | **86.14** | **87.04** | **87.55** |

Table 5: Reaction product prediction results using beam search with different values of beam width on *USPTO-15k*.

Computing all configurations of $(\xi, u, v, b)^\tau$ jointly is very memory demanding, however. Thus, we decompose the first term as follows:

$$
\begin{aligned}
\log p\left((\xi, u, v, b)^\tau | \mathcal{G}, (\xi, u, v, b)^{0:\tau-1}\right) &= \log p\left(\xi^\tau | \mathcal{G}, (\xi, u, v, b)^{0:\tau-1}\right) + \\
&\quad \log p\left((u, v)^\tau | \xi^\tau, \mathcal{G}, (\xi, u, v, b)^{0:\tau-1}\right) + \\
&\quad \log\left(b^\tau | (\xi, u, v)^\tau, \mathcal{G}, (\xi, u, v, b)^{0:\tau-1}\right)
\end{aligned}
$$

At step $\tau$, we do beam search for the signal $\xi^\tau$, then the atom pair $(u, v)^\tau$ and finally the bond type $b^\tau$. Algorithm 1 describes beam search in detail. Some notable technicalities are:

- We only do beam search for $(u, v)$ and $b$ if the prediction is ongoing, i.e., when $\xi^\tau = 1$. To keep track of this, we use a boolean vector $C$ of length $N$ with $C^0$ is initialized to be all true.
- To avoid beam search favoring short sequences, we normalize the log probability scores over sequence lengths. This is shown in lines 10, 17, 32 and 47

### BEAM WIDTH ANALYSIS

Table 5 reports how beam width affects the decoding performance on the *USPTO-15k* dataset. Surprisingly, the top-1 accuracy in case of beam width[3] of 1 is higher than the those when beam widths range from 2 to 15. It means that large beam width is not always good in our situation. However, at beam width of 20, our beam search achieves the best results for different values of $k$. Thus, we set the beam width to 20 in subsequent experiments.

### A.5 USING REAGENT INFORMATION EXPLICITLY

As can be seen from Table 6, reagent molecules account for about a half of the input molecules on average and 60-80% of all reactions containing reagents. It suggests that the proper use of reagent information will lead to better prediction. In our model, before computing the scores for all atom pairs, we append to the representation vector of every atom a binary scalar indicating whether this atom comes from a reagent molecule or not. Then, at the top-$K$ atom pair selection step, we also exclude all atom pairs that have either atoms belong to a reagent molecule. The improvement in prediction accuracy on the validation set of *USPTO-15k* is shown in Fig. 4.

### A.6 COMPARISON WITH ELECTRO

**In method**   Both GTPN and ELECTRO (Bradshaw et al., 2018) are able to explain the mechanism behind a reaction. ELECTRO regards a reaction as an ordered sequence that alternates between removing and adding a single bond. Our model, on the other hand, assumes no specific order of transformations as well as the amount of valences that a bond can change. Thus, our model is more generic than ELECTRO and can cover a much larger set of reactions.

---

[3]Note that beam search with beam width = 1 is different from greedy search as in beam search, as we model the whole sequence probability.

---

**Algorithm 1** Reaction triple prediction using beam search.

---

**Input:** A multi-graph $\mathcal{G}$ consisting of reactant and reagent molecules, number of bond types $E$, max prediction steps $T$, beam width $N$

1: $P^0 = [(-1, -1, -1, -1), ...]$          ▷*The best $N$ subsequences of $(\xi, u, v, b)$*
2: $S^0 = [0, ...]$      ▷*The length-normalized log joint probabilities of the best $N$ subsequences*
3: $C^0 = [\text{True}, ...]$      ▷*The continuation indicator of the best $N$ subsequences*

4: Perform $L$ steps of message passing for all nodes using Eq. (1)
5: $\boldsymbol{x}_i^0 = \boldsymbol{x}_i \ \forall i \in \mathcal{V}$      ▷*The initial states of all nodes before decoding*
6: $\mathcal{N}^0(i) = \mathcal{N}(i) \ \forall i \in \mathcal{V}$      ▷*The initial neighbor set of all nodes before decoding*
7: $\boldsymbol{h}^0$ is loaded from the saved model      ▷*The initial RNN hidden state before decoding*

8: **for** $\tau$ from 1 to $T$ **do**
9:     Find the top $K$ atom pairs $\left\{ (u_k, v_k)^\tau \mid k = \overline{1, K} \right\}$ using Eqs. (4,5)

10:     $S^{\tau-1;0} = S^{\tau-1} \times \frac{\tau-1}{\tau}$      ▷*Superscript $0$ denotes the sub-step $0$*
11:     $P^{\tau-1;0} = P^{\tau-1}; C^{\tau-1;0} = C^{\tau-1}$

12:     *Beam search for continuation signals*

13:     ———————————————
14:     $R^{\text{signal}} = \emptyset$      ▷*Stores the log joint probabilities for $N \times 2$ possible signals*
15:     **for** $n$ from 1 to $N$ **do**
16:         Compute $p\left(\xi^\tau \mid P_n^{\tau-1;0}\right)$ using Eq. (6)
17:         Add $C^{\tau-1;0} \times \frac{1}{\tau} \log p\left(\xi^\tau = \delta \mid P_n^{\tau-1;0}\right) + S_n^{\tau-1;0}$ to $R^{\text{signal}}$ for $\delta \in \{\text{True}, \text{False}\}$
18:     **end for**
19:     Sort $R^{\text{signal}}$ in descending order
20:     $S^{\tau-1;1} = R_{0:N}^{\text{signal}}$
21:     $\bar{\xi}^\tau \equiv$ output signal of $N$ beams in $R_{0:N}^{\text{signal}}$
22:     $I^{\tau-1;1} \equiv$ indices of $N$ beams in $R_{0:N}^{\text{signal}}$
23:     $P^{\tau-1;1} = \text{extract}\left(P^{\tau-1;0}, I^{\tau-1;1}\right)$
24:     $C^{\tau-1;1} = \text{extract}\left(C^{\tau-1;0}, I^{\tau-1;1}\right)$
25:     $C_n^{\tau-1;1} = C_n^{\tau-1;1} \wedge \xi_n^\tau \ \forall n \in \overline{1, N}$
26:     ———————————————

27:     *Beam search for atom pairs*

28:     ———————————————
29:     $R^{\text{atom pair}} = \emptyset$      ▷*Stores the log joint probabilities for $N \times K$ possible atom pairs*
30:     **for** $n$ from 1 to $N$ **do**
31:         Compute $p\left((u, v)^\tau \mid \bar{\xi}_n^\tau, P_n^{\tau-1;1}\right)$ using Eq. (7)
32:         Add $C^{\tau-1;1} \times \frac{1}{\tau} \log p\left((u, v)_k^\tau \mid \xi_n^\tau, P_n^{\tau-1;1}\right) + S_n^{\tau-1;1}$ to $R^{\text{atom pair}} \ \forall k \in \overline{1, K}$
33:     **end for**
34:     Sort $R^{\text{atom pair}}$ in descending order
35:     $S^{\tau-1;2} = R_{0:N}^{\text{atom pair}}$
36:     $(\bar{u}, \bar{v})^\tau \equiv$ output atom pair of $N$ beams in $R_{0:N}^{\text{atom pair}}$
37:     $I^{\tau-1;2} \equiv$ indices of $N$ beams in $R_{0:N}^{\text{atom pair}}$
38:     $P^{\tau-1;2} = \text{extract}\left(P^{\tau-1;1}, I^{\tau-1;2}\right)$
39:     $C^{\tau-1;2} = \text{extract}\left(C^{\tau-1;1}, I^{\tau-1;2}\right)$
40:     $\bar{\xi}^\tau = \text{extract}\left(\bar{\xi}^\tau, I^{\tau-1;2}\right)$
41:     ———————————————

42:     *Beam search for bonds*

43:     ———————————————
44:     $R^{\text{bond}} = \emptyset$      ▷*Stores the log joint probabilities for $N \times B$ possible bonds*

---

**Algorithm 2** Reaction triple prediction using beam search (cont.)

---

45:     **for** $n$ from 1 to $N$ **do**
46:         Compute $p\left(b^\tau \mid (\bar{\xi}, \bar{u}, \bar{v})_n^\tau, P_n^{\tau-1}\right)$ using Eq. (8)
47:         Add $C^{\tau-1;2} \times \frac{1}{\tau} \log p\left(b^\tau = \beta \mid (\bar{\xi}, \bar{u}, \bar{v}), P_n^{\tau-1}\right) + S_b^{\tau-1}$ to $R^{\text{bond}} \; \forall \beta \in \overline{1, B}$
48:     **end for**
49:     Sort $R^{\text{bond}}$ in descending order
50:     $S^{\tau-1;3} = R_{0:N}^{\text{bond}}$
51:     $\bar{b}^\tau \equiv$ output bond of $N$ beams in $R_{0:N}^{\text{bond}}$
52:     $I^{\tau-1;3} \equiv$ indices of $N$ beams in $R_{0:N}^{\text{bond}}$
53:     $P^{\tau-1;3} = \text{extract}\left(P^{\tau-1;2}, I^{\tau-1;3}\right)$
54:     $C^{\tau-1;3} = \text{extract}\left(C^{\tau-1;2}, I^{\tau-1;3}\right)$
55:     $\bar{\xi}^\tau = \text{extract}\left(\bar{\xi}^\tau, I^{\tau-1;3}\right)$
56:     $(\bar{u}, \bar{v})^\tau = \text{extract}\left((\bar{u}, \bar{v})^\tau, I^{\tau-1;3}\right)$
57:     ______________

58:     $S^\tau = S^{\tau-1;3}; C^\tau = C^{\tau-1;3}$
59:     $P_n^\tau = \text{append}\left(P_n^{\tau-1;3}, (\bar{\xi}, \bar{u}, \bar{v}, \bar{b})_n^\tau\right)$

60:     **for** $n$ from 1 to $N$ **do**
61:         Update the $\mathcal{N}^\tau(\bar{u}_n)$ and $\mathcal{N}^\tau(\bar{v}_n)$ for all $n = \overline{1, N}$ using Eqs. (26,27)
62:         Update $\boldsymbol{x}_{\bar{u}_n}^\tau$ and $\boldsymbol{x}_{\bar{v}_n}^\tau$ using Eq. (1)
63:         Perform $m$ steps of message passing for all nodes in the graph
64:         Update $\boldsymbol{h}^\tau$ using Eq. (25)
65:     **end for**
66: **end for**
**Output:** $P^T, S^T$

---

| Dataset | | %reactions containing reagents | %reagents over input molecules |
|---------|------|-------|-------|
| *USPTO-15k* | train | 63.1% | 41.3% |
| | valid | 65.3% | 42.3% |
| | test | 63.6% | 40.9% |
| *USPTO* | train | 79.7% | 54.0% |
| | valid | 80.0% | 54.4% |
| | test | 79.9% | 54.2% |

Table 6: Proportion of reactions containing reagents and proportion of reagents over input molecules on *USPTO-15k* and *USPTO*.

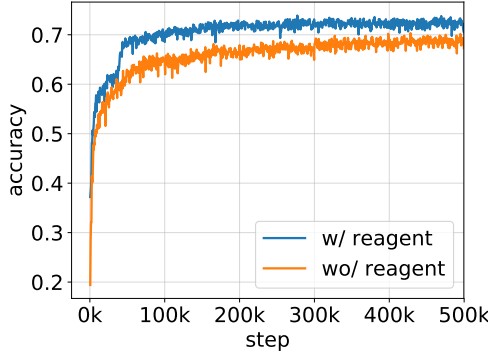

Figure 4: Learning curves of our model with and without using reagent information explicitly on *USPTO-15k*.

| Model | Processed USPTO | | |
|---|---|---|---|
| | P@1 | P@3 | P@5 |
| WLDN (Jin et al., 2017) | 84.0 | 91.1 | 92.3 |
| ELECTRO (Bradshaw et al., 2018) | 87.0 | **94.5** | **95.9** |
| GTPN$^{\diamond\spadesuit\clubsuit}$ | **87.35** | 90.22 | 90.68 |

Table 7: Results for the reaction prediction task. *P@k* is the precision at *k*. Best results are highlighted in bold. Meanings of markers in our model: $\diamond$: With beam search (beam width = 20), $\spadesuit$: Invalid product removal, $\clubsuit$: Duplicate product removal.

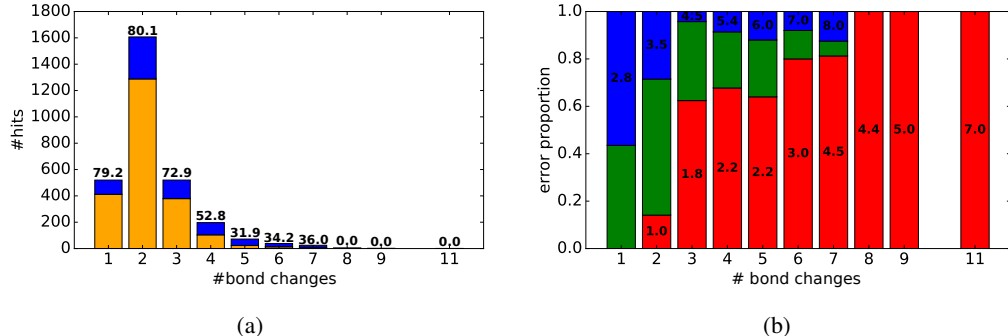

(a)                                              (b)

Figure 5: Performance with respect to different numbers of bond changes. (a) Top-1 accuracy. (b) Errors grouped by length. In (a), **blue**: all reactions having that sequence length; **orange**: correct predicted reactions. In (b), **red**: the predicted sequence is shorter (than the groundtruth sequence); **green**: the predicted and the groundtruth have the same length; **blue**: the predicted sequence is longer; number indidate the average length.

**In performance**    To do a fair comparison with ELECTRO (Bradshaw et al., 2018), we follow their procedure described in the paper to prepare a new test set that contains only reactions with linear chain topology and single-valence bond changes. It results in 29,808 reactions, close to the reported number of 29,360 in (Bradshaw et al., 2018). We reuse our old model (see Section 3.4) trained on the original *USPTO* dataset. We also use beam search decoding and post-processing as similar to (Bradshaw et al., 2018). From Table 7, we see that GTPN achieves the highest top-1 accuracy of 87.35%, outperforming ELECTRO and WLDN by 0.35% and 3%, respectively. For the top-3 and top-5 accuracies, our model, however, does worse than the other two. Especially, while both ELECTRO and WLDN have big jumps from *P@1* to *P@3* with about 7% improvement, GTPN only has 3% increase. We conjecture that this problem mainly comes from the fact that GTPN was not optimized on the compatible training and validation sets.

## A.7    ERROR ANALYSIS

In this section, we analyze several error types that our model makes during prediction. All the results below are computed on the *USPTO-15k* dataset by using beam search decoding with the beam width $N = 20$ and no post-processing.

**Errors grouped by number of bond changes**    Fig. 5 shows the top-1 accuracies for reactions with different number of bond changes. Our model performs poorly on reactions with many bond changes. However, those kinds of reactions only accounts for a small proportion in the dataset. From Fig. 5b, we see that the lengths of the error sequences tends to be shorter than the lengths of the groundtruth sequences.

**Errors caused by signal/atom pair/bond type**    We define a sub-action causing error as the first sub-action that our model makes a wrong decision. In Fig. 6a, we plot the the proportion of errors with respect to the three kinds of sub-actions. Clearly, atom pair prediction causes the most errors

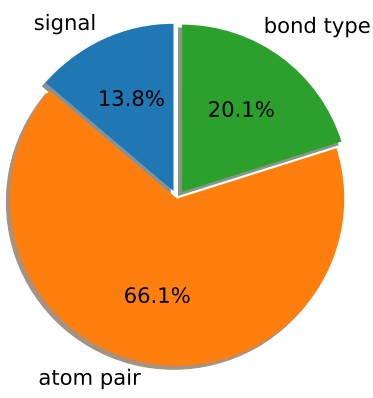
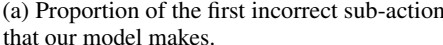
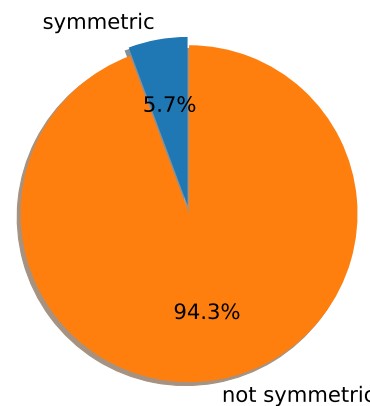

(a) Proportion of the first incorrect sub-action that our model makes.

(b) Proportion of the incorrect top-1 products that have similar structure to the groundtruth products.

Figure 6: Errors grouped by the first incorrect sub-actions (a), and errors caused by symmetric structures (b).

(nearly two third). This makes sense because this sub-action is harder than signal prediction and bond type prediction. Therefore, more effort should be put on improving the prediction of atom pairs.

**Errors caused by symmetry**    There exists cases in which different sequences of bond changes can result in the same products due to symmetric graph structures. Errors caused by symmetry account for 5.7% of the top-1 errors on the *USPTO-15k* dataset as shown in Fig. 6b. For better understanding, we provide a short list of wrong reaction triple predictions caused by symmetry in Fig. 7. In this list, the top-1 products (along the second column) are incorrect while the top-2 products (along the third column) are correct though both have the same probability.

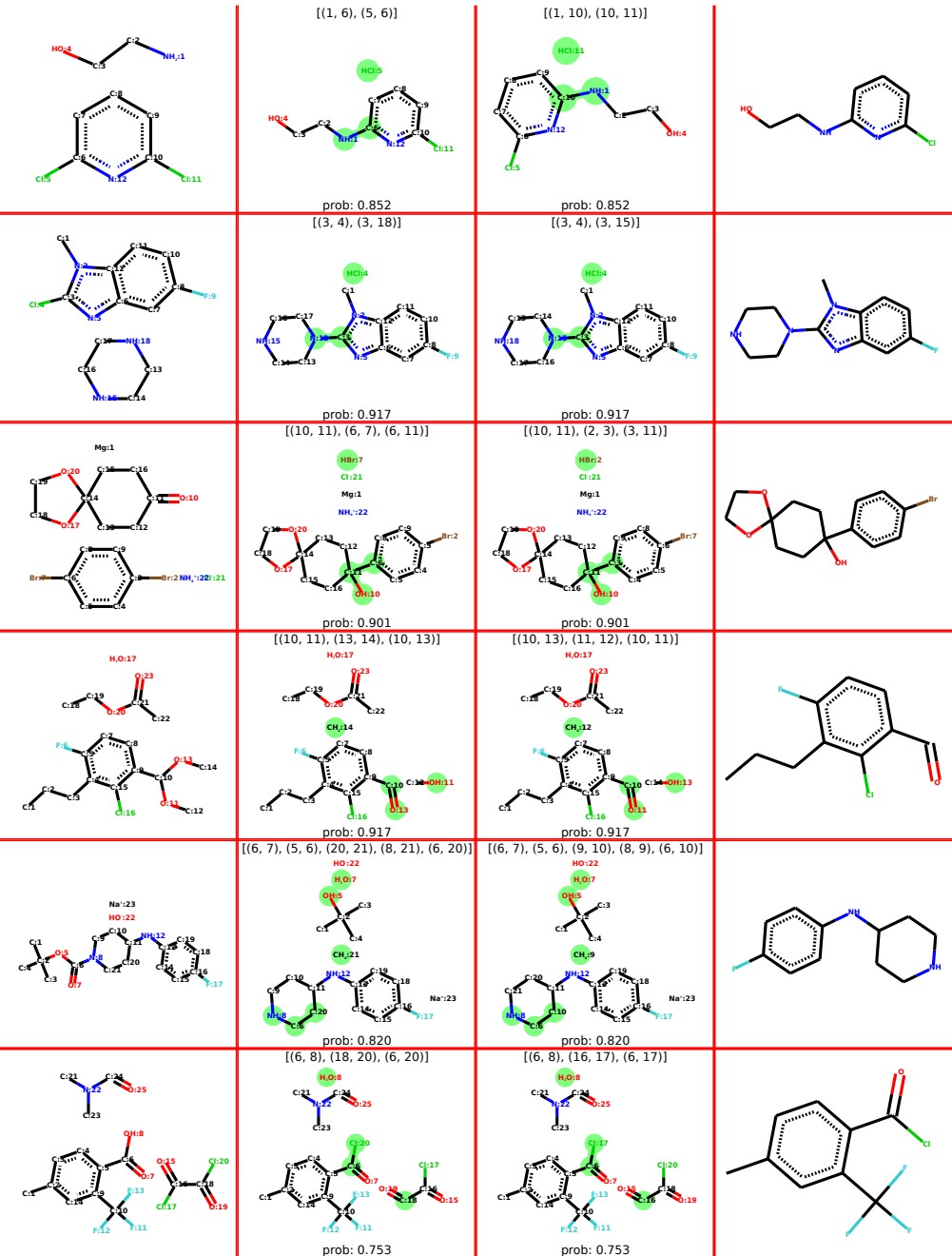

Figure 7: Visualization of some reactions that cause multiple products with symmetric structures. Each row corresponds to a reaction. The columns, from left to right, show: i) reactant and reagent molecules, ii) **incorrect** top-1 product molecules, iii) **correct** top-2 product molecules, and iv) major groundtruth product molecules. All atoms in the first three columns are labeled with their atom map numbers. For the top-1 and top-2 products, we highlight the predicted reaction triples in green and provide the probability of the predicted sequence at the bottom.

