# OpenReview forum: "GRAPH TRANSFORMATION POLICY NETWORK FOR CHEMICAL REACTION PREDICTION"
_ICLR.cc/2019/Conference_

### Official Review · AnonReviewer3 · 2018-10-31

**Rating:** 5
**Confidence:** 4

**Review:**

The paper provides a new system that combines a number of neural networks to predict chemical reactions. The paper brings together a number of interesting methods to create a system that outperforms the state of the art.

Good about this paper:
 - reported performance: the authors report a small but very consistent performance improvement.
 - the authors propose an approach that puts together many pieces to become an effective approach to chemical reaction prediction.
 -

Problematic with this paper
 - this paper is impossible to understand if you only refer to the ten pages of content. There are at least 5 pointers in the paper where the authors refer to the appendix for details. Details in many of these cases are necessary to even understand what is really being done:  p3: rewards p4: message passing functions, p5: updating states, p9: training details. Further, The paper has some details that are unnecessary - e.g. the discussion of global vs. local network on p4 - this could go into the appendix (or be dropped entirely)

 - the model uses a step-wise reward in the training procedure (p3) -> positive reward for each correct subaction. It is not clear from the paper whether the model requires this at test time too (which should not be available). It's not clear what the authors do in testing. I feel that a clean RL protocol would only use rewards during training that are also available in testing (and a final reward)

 - eq 7: given there is an expontential in the probability - how often will the sampling not pick the top candidate? feels like it will mostly pick the top candidate.

 - eq 9: it's unclear what this would do if the same pair of atoms is chosen twice (or more often)

 - the results presented in table 3: it appears that GTPN alone (and with beam search) is worse than the previous state of the art. only the various post processing steps make it better than the previous methods. It's not clear whether the state of the art methods in the table use similar postprocessing steps or whether they would also improve their results if the same postprocessing steps were applied.


minor stuff:
p2: Therefore, one can view GTPN as RL -> I don't think there is a causality. Just drop "Therefore"
p2: standard RL loss -> what is that?
eq. 2: interestin gchoice to add the vectors - wouldn't it be easier to just concatenate?
p4: what does "NULL" mean? how is this encoded?
p4 bottom: this is quite uncommon notation for me. Not a blocker but took me a while to parse and decrypt.
p5: how are the coefficients tuned?

---

> ### Author Response · Authors · 2018-11-09
> **Reply to Reviewer 3**
>
> Thank you for your comments and suggestions. We would like to address your concerns as follows:
>
> 1) "This paper is impossible to understand if you ... p9: training details."
>
> Thank you for your suggestions. We agree with the reviewer that it is not easy to fully understand our paper in just the official ten pages even though we have tried our best to present the most important things within the page limit. Here, we would like to explain why we put the sections you mentioned in the appendix:
> - We described the reward we used in our method with a separate paragraph in the Sec 2.1 and we think it is quite clear to understand. We also provide more details about the reward values in the appendix A.3.
> - Graph message passing neural networks are quite common in the machine learning literature and have been introduced many times in other works so we kept them in the appendix. The idea of message passing is to provide every node in a graph with local structure information.
> - Because updating states is not the main focus when presenting our method so we put it in the appendix.
> - For training, the loss functions are important for building our model so we described and explained them carefully, while other less important details like learning rate or batch size are put in the appendix.
>
> 2) "The discussion of global vs. local network on p4. This could go into the appendix (or be dropped entirely).":
>
> The local/global network is part of the “node pair prediction network” component in our model so we cannot drop it entirely. Moreover, the two concepts of “local” and “global” come from the work of Jin et. al., 2017 so we feel responsible to make it clear. It seems that this discussion does not consume much space in our paper. We hope the reviewer could understand.
>
> 3) "The model uses a step-wise reward ... in testing (and a final reward)."
>
> The delayed (or final) reward is the reward given to the model when comparing the generated product with the ground truth product and it can be available during both training and testing.
>
> During training, if at each sub-step, the predicted sub-action is correct, we also give our model an immediate positive reward, otherwise, an immediate negative reward. Using intermediate reward during training is a common practice in RL to improve the performance.
>
> During testing, intermediate rewards are not available. In fact, in testing, we do not care about the intermediate rewards or the final reward. We evaluate our model using the Precision@k metric. For a reaction, if the top k candidate products contain the ground truth product, we give it one point.
>
> 4) "Eq 7: given there is an exponential in the probability - how often will the sampling not pick the top candidate? Feels like it will mostly pick the top candidate.":
>
> In Eq. 7, the probability of selecting an atom pair is the softmax over the top K atom pairs (computed by the node pair prediction network (NPPN)). This probability affects how often the top candidate will be picked up. During the early stage of learning, our model may sample an atom pair from the top K randomly. If it chooses the wrong one, it will be given a negative reward, otherwise a positive reward. Over time, our model will assign higher probability to the correct reaction atom pair and sample it more frequently. This is when our algorithm comes to convergence. However, there may be the case that our model converges too fast and may not do enough exploration. In this situation, a better exploration strategy may be needed. Although we did not observe this phenomenon in our experiment, we will note this idea for our future work. We thank the reviewer for the suggestion.
>
> 5) "Eq 9: it is unclear what this would do if the same pair of atoms is chosen twice (or more often).":
>
> We really thank the reviewer for pointing this out. We did not mention this carefully in our paper and we have fixed this in our revised version. In our method, we do not allow an atom pair to be chosen twice. Once an atom pair is predicted at a step, its position in the link matrix will be masked so that it does not appear in the top K atom pairs at the following steps, and our model will not pick it again. This is the common practice when predicting a set: we remove an element once we have predicted it and predict the remaining until we reach a stop symbol/signal.
>
> 6) "The results presented in ... the same post-processing steps were applied":
>
> The reviewer’s observation is right. Without post-processing, we cannot beat previous state-of-the-art methods and we mentioned it in our paper. Like our method, other state-of-the-art methods also use post-processing (e.g. Bradshaw et. al., 2018) or handcrafted reaction rules/templates (e.g. Jin et. al., 2017). Their reported results have already included these steps.

---

> > ### Comment · AnonReviewer3 · 2018-12-05
> > **Thanks for the responses.**
> >
> > Although the authors have addressed some concerns, I am now even more concerned that this paper is way too long to be fairly compared to other submissions that stick to the guideline of using ten pages.
> >
> > This also shows in the very long review responses by the authors.

---

> > > ### Author Response · Authors · 2018-12-07
> > > **Thanks for your comment**
> > >
> > > In our paper, everything we say is related to our work so we hope the reviewer could consider the length of the paper including supplemental material as reflecting the volume of the work we did.
> > >
> > > Although we tried our best to present our problem and solution clearly as we could, sometimes, it still causes the reviewers’ misinterpretation. Through our answers, we would like to clarify it again. We do not mind providing a detailed answer to every question from the reviewers and seems that it is a common practice in Open Review as other authors also do the same as us. Thus, we encourage Reviewer 3 to give questions if you have any more concerns and we are pleased to address them in an open-minded manner.

---

> ### Author Response · Authors · 2018-11-09
> **Reply to Reviewer 3 (cont.)**
>
> 7) "p2: Therefore, one can view GTPN as RL -> I don’t think there is a causality. Just drop “Therefore”.":
>
> We agree with the reviewer and we have removed this word from our paper.
>
> 8) "p2: standard RL loss -> what is that?":
>
> “standard RL loss” is the policy gradient loss that we later described in Sect 2.4. We have replaced this term with the appropriate term. We thank the reviewer for pointing this out.
>
> 9) "Eq. 2: interesting choice to add the vectors - wouldn’t it be easier to just concatenate?":
>
> Concatenation will break the symmetry in the score of an atom pair (if you switch 2 atoms). That is the reason why we add 2 vectors. This was previously used in Jin et. al., 2017
>
> 10) "p4: what does “NULL” mean? how is this encoded?":
>
> “NULL” is a special bond type to indicate that there is no bond between 2 atoms. We encode it like other bond types by using an embedding matrix. Please refer to the Appendix A.3 for more details.
>
> 11) "p4 bottom: this is quite uncommon notation for me. Not a blocker but took me a while to parse and decrypt":
>
> We are sorry for causing difficulty in the reviewer’s interpretation. Actually, we use the notation to represent a set with K elements indexed from 1 to K.
>
> 12) "p5: how are the coefficients tuned?":
>
> Actually, we did not spend much effort to tune the coefficients of the losses. Instead, finding the suitable formulas for the “over-length” loss and the “in top K” loss and implementing them are more time-consuming.
>
> [Jin et. al., 2017] Predicting Organic Reaction Outcomes with Weisfeiler-Lehman Networks

---

### Official Review · AnonReviewer2 · 2018-11-02
**Nice application. Some edits are needed.**

**Rating:** 6
**Confidence:** 5

**Review:**

Update:

Score increased.

___________________________________

Original review:

The paper presents an approach to predict the products of chemical reactions, given the reactants and reagents. It works by stepwise predicting the atom pairs that change their bonds in course of reaction, and then adjusting the bonds between them. This can be interpreted as a stepwise graph transformation.

I think this is an interesting applied ML paper with fair results. The presentation is clear and understandable. The experimental setup is reasonable. However, the paper is not ready yet to be accepted in my opinion.

I think a higher score is justified if the authors address the following points:

- Relation to previous work, originality

In contrast to what the authors claim, what is predicted here is not exactly the reaction mechanism, but an implementation of the principle of minimal chemical distance, which was already described by Ugi and coworkers in 1980 [see Jochum, Gasteiger, Ugi, The Principle of Minimum Chemical Distance Angew. Chem. Int. Ed. Engl. 1980, 19, p 495-505].
The “insight” the authors have about treating reagents and reactants jointly is Organic Chemistry 101, and that reactions are stepwise graph transformations was also reported by Ugi et al, however, already in 1979! [Ugi, et al. "New applications of computers in chemistry." Angewandte Chemie International Edition in English 18.2 (1979): 111-123. ] I assume the authors were not aware of these papers, but now they are, so this needs to be modified accordingly, and these papers need to be referred to in the introduction.



- Questions:

The authors suggest that graph neural networks are more generic that so-called heuristic features (fingerprints) – which, as Duvenaud et al have elaborated, can be interpreted just as graph neural networks themselves – with fixed weights. Also, there are results by the Hochreiter group which show that graph neural networks perform worse that classical chemical features under rigorous testing { DOI: 10.1039/C8SC00148K } Do the authors think their models could also improve if they used the classical fingerprints?


Is the GRU really needed to encode the past bond changes? What happens if you remove it?

The statement that the method has the advantage of not relying on handcrafted reaction templates is somewhat overselling, because instead it uses a handcrafted complex neural network architecture. How complicated is it to train the network? If you remove some of the “tricks” of shaping the loss function, does it still train well?


To what degree is the ranking of the different models just a matter of hyperparameter tuning or different architectures? If you used a different graph neural net instead of an MPNN on top of your GTPN method, what would you expect? Are the differences between the models significant?


During prediction, you apply a flag to the starting molecules if they are a reagent or reactant. How do you know upfront what a reagent or reactant is during inference?

On page 5 and 7, you speak of the correct sequence of reaction triples (which implies an ordering), even though earlier you claim the algorithm is order-invariant? Where do you get the ground truth labels from? I assume these are already annotated in the data.


In the appendix, please replace the pie chart with a bar chart.

- Language:
I would suggest the authors to adapt the language of their paper towards a more academic tone. Science is not a sports competition of getting slightly higher numbers in benchmarks, but rather about providing insights and explanations. Words like “beating” or “record” are locker room talk, and to be avoided.

---

> ### Author Response · Authors · 2018-11-09
> **Reply to Reviewer 2**
>
> Thank you for your detailed comments and suggestions. We would like to address your concerns as follows:
>
> 1) "Relation to previous work, originality":
>
> We think the originality of our method was clearly presented in our Introduction. In this answer, we would like to highlight again the main difference between our method and other methods. Most previous methods solved the chemical reaction prediction problem by “doing multi-stage prediction with the aid of reaction rules/templates”. The most recent work of Bradshaw et. al., 2018 built a method that can learn end-to-end but they made many assumptions and so their method can only handle some special types of reactions. In our work, we try to address those existing problems of other methods and build a “generic” model that i) learns end-to-end from raw data without using reaction rules/templates ii) does not make any assumptions about the types of reactions, and iii) interprets the reaction process. In the Introduction section, we also discussed major difficulties when building such a generic model, one of them is “handling a large complex and non-differentiable space of sets of reaction triples”. In our Method, we carefully presented different techniques to deal with this difficulties.
>
> 2) "In contrast to what the authors claim, ... need to be referred to in the introduction.":
>
> We thank the reviewer for this insightful comment. We used the definition of the term “reaction mechanism” from Chem LibreTexts which is stated as “the step-by-step description of what occurs on a molecular level in chemical reactions”. Honestly, chemistry is not our expertise so we might misuse the term “reaction mechanism” in our paper. What we actually mean is “a set (or unordered sequence) of bond changes” and we have modified our expression in the revised version along this way to make it clear.
>
> We also thank the reviewer for introducing the two papers “Principle of Minimum Chemical Distance” of Jochum et. al., 1980 and “New applications of computers in chemistry” of Ugi et. al., 1979. We are happy to cite them in our revised version. Upon studying the papers carefully we found that these papers discuss very different concepts such as “minimum chemical distance” and “isomeric ensemble” which are not closely related to the problem of predicting reaction products.
>
> We agree that the “insight” to represent a reaction as a sequence of graph transformations is Chemistry 101. However, we would like to clarify that our main contribution in this work is not about discovering that “insight” but about building a generic end-to-end machine learning model which realizes that “insight”. From the difficulties in building such a generic model that we have addressed in our paper and the details of the solution we presented, we really hope that our effort could be fairly recognized.
>
> 3) "The authors suggest that graph neural ... the classical fingerprints?":
>
> We would like to clarify the point here: In our paper, we did not claim that GNNs are more generic than classical fingerprints. Instead, we said that our entire model (including GNN, node pair prediction network, and policy network) is a generic architecture for the reaction prediction problem.
>
> We agree with the reviewer that graph neural networks (GNNs), Morgan fingerprints and even SMILES strings are widely used in Chemical Informatics. Different methods provide different ways to represent molecules. And many other papers, including the work that the reviewer suggested, have shown that sometimes using fingerprints or SMILESs can lead to better results than using GNNs. However, in our method, modeling reactant/reagent molecules is only one step. The more important steps are: i) predicting reaction atom pairs and their new bonds, and ii) updating new change in the structure of the molecules. To do this, we need the information/state of every atom available. This is the reason why we use GNNs because it supports node embedding. Fingerprints and SMILESs, on the other hand, represent molecules as a whole so they are not suitable to our current architecture. In the future, if we can find a way to integrate the classical fingerprints into our model, we will definitely make a comparison with GNNs.
>
> 4) "Is the GRU really needed to encode the past bond changes? What happens if you remove it?":
>
> Before arriving at the current architecture, we had built a simpler version without GRU. It converged slower and did not give the result we expected. Then we added GRU and it worked better. The intuition is that using GRU, the model can remember previously predicted reaction triples so it is more likely to select the correct pair and its new bond. If it is possible, we would like to upgrade our GRU to an external memory. We think it will not introduce much computational cost since the number of bond changes is small.

---

> > ### Comment · AnonReviewer2 · 2018-12-07
> > **thanks for the update**
> >
> > I want to thank the authors for the update, and have increased the score.

---

> > > ### Author Response · Authors · 2018-12-08
> > > **Thank you for your consideration**
> > >
> > > We thank Reviewer 2 for your great consideration.

---

> ### Author Response · Authors · 2018-11-09
> **Reply to Reviewer 2 (cont.)**
>
> 5) "The statement that the method has the advantage of not relying on handcrafted reaction templates is somewhat overselling, because instead it uses a handcrafted complex neural network architecture.":
>
> Honestly, we do not think that methods using handcrafted reaction templates are not advanced because they show a great combination of expertise in both machine learning and chemistry. However, from deep learning researchers’ point of view, we thought that we can totally automate the reaction prediction process without using reaction rules/templates and we proved this statement with our model. That is why in our paper, we did not say that our method is more advanced than other methods but we say our method is “generic” for the reaction prediction problem. We also carefully explain the meaning of the term “generic” in our Introduction.
>
> In fact, our model is not very complex as it may seem. All three components are fundamental and can be easily understood: a graph neural network for modeling input molecules, a node pair prediction network for choosing a possible set of reaction node pairs, and a policy network for predicting a specific reaction triple. We could not create a well-performed generic model if we lack one of these components.
>
> 6) "How complicated is it to train the network?":
>
> Training our model is also not complex. The evidence is that we implemented our model using Tensorflow and trained it entirely end-to-end on a single GPU GTX1080Ti 12GB. All the formulas to build our model are provided in the paper. However, to really get insights and control the training process requires practical experiences about graph neural networks and policy gradient.
>
> 7) "If you remove some of the “tricks” of shaping the loss function, does it still train well?":
>
> The losses that we introduced in our paper play important roles in the performance in both time and accuracy and should not be removed. For the reason why they are important, we explained carefully in our paper because we really want to make it clear to the readers. We hope the reviewer could understand that we did not come up with those formulas in one day but after many trials and errors. We think this is a good practice in research since research is an iterative process of trials and adjustments.
>
> 8) "To what degree is the ranking of the different models just a matter of hyper-parameter tuning or different architectures?":
>
> We would like to say that our major goal when doing this work is not to compete with other available methods but to create a generic model that can learn how to predict reactions directly from raw data in an end-to-end manner without the aid of reaction rules. At the time we started our work, almost all methods do multi-stage prediction and use reaction templates, which did not satisfy us in term of modeling. That is why we take this challenge. Like other research, we also started with simple models first and only made things more complex if they were reasonable and led to better results. Building a simple model that learns end-to-end for this task is not difficult (and we actually did it). However, building a model that can i) learn end-to-end, ii) be applicable to various types of reactions, and iii) give good results is not easy at all. Both the architecture and the hyper-parameters we used are crucial in the performance of our method. In our paper, we also give a lot of details about them so we recommend the reviewer to refer to that. However, we are not fans of hyper-parameter tuning. That is the reason why in our paper (the Appendix A.3), we say that we used the optimal hyper-parameters found by training on the dataset USPSTO-15k to train our model on USPTO without fine-tuning.
>
> 9) "If you used a different graph neural net instead of an MPNN on top of your GTPN method, what would you expect? Are the differences between the models significant?":
>
> Since different variants of Message Passing Neural Networks (MPNNs) share the same mechanism, we think their performances are comparable. It is also reflected in Table. 2 of our paper where we compare our MPNNs with the other two variants WLNs and CLNs.

---

> ### Author Response · Authors · 2018-11-09
> **Reply to Reviewer 2 (cont. 2)**
>
> 10) "During prediction, you apply a flag to the starting molecules if they are a reagent or reactant. How do you know upfront what a reagent or reactant is during inference?":
>
> We agree that during prediction, there is no way to know unless you explicitly indicate which input molecules are reactants and which are reagents. In our paper, we assumed that we know which molecules are reactants and which are reagents in advance. This information helps our model learn faster. Our model can still perform quite well when no reagent information is available. However, to reach the performance of the model with reagent information requires longer training time. In Appendix A5, we compared our model with and without using reagent information and emphasized the importance of reagent information. We recommend the reviewer to check this.
>
> 11) "On page 5 and 7, you speak of the correct sequence of reaction triples (which implies an ordering), even though earlier you claim the algorithm is order-invariant?":
>
> Actually, our model predicts a set of reaction triples. However, to technically handle sets with variable length, we have to treat them as sequences. In our paper, we used the term “order-free” (instead of “order-invariant”) which should be understood like that: We do not force our model to predict reaction triples according to any specific order. Instead, our model automatically discovers the best order itself. This approach has been previously studied for set (Welleck et. al., 2017) and multi-label (Wang et. al., 2016) prediction problems.
>
> 12) "Where do you get the ground truth labels from? I assume these are already annotated in the data.":
>
> The ground-truth sets of reaction triples that we use for training are not already annotated but can be easily inferred from the data. For the USPTO dataset, given the reactant/reagent molecules and the product molecules whose atoms are uniquely identified, we find reaction atom pairs and their corresponding new bonds by comparing each atom pair in the reactant/reagent molecules with the counterpart in the product molecules. This extraction process has been discussed previously in the work of Jin et. al., 2017 so we did not describe it carefully in our paper. We thank the reviewer for pointing this out. We have provided details in our revised version.
>
> 13) "In the appendix, please replace the pie chart with a bar chart.":
>
> We thought the pie charts are suitable for representing proportions, but we are happy to change as your advice. Could you please advise any particular reason for this request?
>
> 14) "I would suggest the authors to adapt the language ... and to be avoided":
>
> We thank the reviewer for this useful advice. We have replaced these words with more suitable words in our revised version. We also share the same view as the reviewer about the main purpose of our work and other research works in general, which is to provide insights instead of simply competing with other methods. That is the reason why we try to present as much detail about the model architecture and its settings as possible in our paper.
>
> [Wang et. al., 2016] CNN-RNN: A Unified Framework for Multi-label Image Classification
> [Welleck et. al., 2017] Loss Functions for Multiset Prediction
> [Jin et. al., 2017] Predicting Organic Reaction Outcomes with Weisfeiler-Lehman Networks
> [Bradshaw et. al., 2018] Predicting Electron Paths

---

### Official Review · AnonReviewer1 · 2018-11-03
**Somewhat incremental results and an important point why we can apply RL in this setup remains unclear**

**Rating:** 5
**Confidence:** 4

**Review:**

Summary:
This paper develops a novel method for predicting organic chemical reactions, in particular, final product prediction from given reactants. Organic molecules consist of covalent bonds, and hence organic reactions can be regarded as an alternating multistep series of bond breaking and bond forming (i.e. qualitative understanding as "reaction mechanisms" against quantum chemical calculations). The developed method aims to predict this series of bond changes through a form of reinforcement learning guided with neural networks. In this sense, the setup and formulation seem largely inherited from cited previous papers by Bradshaw et al, 2018 or Kayala & Baldi, 2011 (though they are not used any RL formulation). Organic compounds at each elementary step are represented as molecular graphs, and reactions are thus a series of graph transformations. Each bond change can be considered as "action" at that state to form the next states to head for the final product. The method itself seems quite natural: States transitions are shared by an RNN and the hidden states and observations (molecular graphs at each step, and bond changes as actions) are used to learn "policy" and "state transition" for RL via graph neural networks. Atom pairs to lead the bond change are detected from such graph embeddings and state observations through self-attentive architectures. In addition, masking by additional indicator variables is introduced to avoid redundant training as well as determine the termination of the reaction. Experimental evaluations on a standard large benchmark dataset of USPTO show improved prediction performance compared to previous methods of Jin et al, 2017 and Schwaller et al, 2018.

Comment:
- Given that a chemical reaction can be regarded as a multi-step chain of bond breaking and forming, thus the method part seems a quite natural extension of the past effort but also sounds rather incremental even though the performance gain exists.

- The method part is written clearly but the problem setup seems rather unclear. The most unclear point is how the environment for RL can give any reward to each bond change. How we can know each prediction of a bond change is correct or not? Can it be supervised? Does this mean that the training set of reactions has the correct perfect information of these multi-step bond changes?? How did you construct such curated dataset for USPTO? If this is the case, the motivation to go for RL would be more understandable but this part is not explained at all. (Because it is inherited from previous work of Bradshaw et al 2018 or something?? )

- Why this proposed method has no limitation whereas the previous method by Bradshaw et al, 2018 are limited to "linear chain topology" (?). If this method is the direct competitor, what points are important to remove this limitation of the previous method should be clarified more clearly. This method is referred multiple times in the paper, but no direct explanations exist.

Pros:
	- Nice and solid design of proposed "graph transformation policy network"
	- Better prediction performance against previous methods

Cons:
	- Why this extension could break the previous limit (of ELECTRO?) remains unclear
	- The descriptions on the problem setup and the data are unclear. Perfectly curated as chemically correct multi-steps of bond changes are given as training set? How we can know each prediction of a bond change is correct or not?
	- the proposed architecture is nice but somewhat seems incremental. (heuristic combinations of existing techniques of RL and graph neural nets)

---

> ### Author Response · Authors · 2018-11-09
> **Reply to Reviewer 1**
>
> Thank you for your detailed comments and suggestions. We would like to address your concerns as follows:
>
> 1) "Given that a chemical reaction ... the performance gain exists":
> The problem addressed in the paper, “chemical reaction prediction”, has not been solved completely yet. To the best of our knowledge, none of the previous methods is generic in the sense that: i) performing well with various types of reactions by ii) just using raw data without the add of reaction rules/templates, and iii) can be trained end-to-end. Our method is the first method that has all these characteristics at once. In our paper, we also discussed why building such a generic method like ours is challenging. There are multiple issues associated with achieving generality, including: graph modeling, sequence ordering, a complex discrete space of sets of reaction triples, and many other technical issues such as: loss functions, convergence, and performance. We really hope the reviewer could see all the challenges that we experienced and overcame to bring a complete solution to this open problem.
>
> 2) "How the environment ... multi-step bond changes?":
> Like other works about chemical reaction prediction, our method is totally supervised. The input is reactant/reagent molecules and the output is a set of reaction triples. The detailed description of our model's input/output/reward can be found in Sec. 2.1. During training, for each reaction, we have a ground-truth set of reaction triples available. If the current predicted atom pair and its corresponding new bond do not appear in the ground-truth set, they are incorrect and we give our model negative rewards. Otherwise, the model receives positive rewards.
>
> 3) "How did you construct ... Bradshaw et. al. 2018 or something?)":
> To train our model, we need reaction/reagent molecules and a ground-truth set of reaction triples. The USPTO dataset provides reactant/reagent molecules and product molecules whose atoms are identified with unique “atom map numbers”. By comparing every atom pair of the reactants and reagents against that of the products we can find which atom pairs are modified during a reaction and their corresponding old/new bonds. Our described procedure can be found in the work of Jin et. al., 2017, in which the authors also train a network to detect reaction atom pairs. They also provide codes for extracting data at: https://github.com/wengong-jin/nips17-rexgen. Bradshaw et. al., 2018 might do the same thing as us but this does not necessarily mean we follow their work.

---

> ### Author Response · Authors · 2018-11-09
> **Reply to Reviewer 1 (cont.)**
>
> 4) "Why this proposed method ... no direct explanations exist."
> Before answering the question, we would like to say that the work of Bradshaw et. al., 2018 was not publicly available at the time we began our work. We discovered this work later after we had developed the idea about using reinforcement learning (RL) for chemical reaction prediction. During that time, RL had been widely used for drug generation (e.g. You et. al., 2017) but not many papers applied it to chemical reaction prediction. The absence of a proper method for this problem strongly motivated us to do our work.
>
> Our work and the work of Bradshaw et. al., 2018 take the same approach of predicting a set/sequence of bond changes that transform reactants into products. This is normal because the idea is intuitive. However, there are several key differences between our model and theirs. To clarify these differences, we would like to briefly describe their method ELECTRO: It is a recurrent neural network (RNN) which, at each step, takes a graph of intermediate product molecules as input and outputs a particular atom (identified via its index) that has bond added or removed. To apply this model to predict a sequence of bond changes, Bradshaw et. al., 2018 has made several assumptions which are explicitly stated in their paper, especially in Sec 4.2 of the version 1 on Arxiv:
> – To deal with the ordering problem of RNNs (previously studied by Vinyals et. al., 2016), they only consider reactions with linear chain topology. Other reactions with different topologies are discarded. Using “linear chain” reactions, they can define a special type of order that alternates between removing and adding bonds between atom pairs.
> – To know whether to add or remove bond of the predicted atom a_{t} at the current step t, they check whether t is even or odd. If t is even, they remove a bond between a_{t} and a_{t-1}. Otherwise, they add a bond between a_{t} and a_{t-1}.
> – They do not predict the number of valences added to each atom pair. Instead, they explicitly set it to be one. Thus, their method does not support reactions which have more than one valence change e.g. from single to triple bond or from no bond to double bond. All reactions belonging to this type are removed from the USPTO dataset.
>
> In our opinion, their method has interesting properties such as being lightweight and easy to train. But the trade-off is that it only supports specific types of reactions. On the other hand, our ultimate target is to create a generic method that can predict various types of reactions without any assumptions. That is why we chose RL (or more precisely the policy gradient method) since it has been successfully applied for predicting sets (Welleck et. al., 2017) and discrete nodes/links in graphs (You et. al., 2017).
>
> Please understand that we do not consider the work of Bradshaw et. al., 2018 or any other related work as our direct competitors since our goal and model are very different from theirs. The assumptions made in Bradshaw et. al., 2018 are important to keep their model simple enough and we agree with them about that. However, unlike other methods, the model of Bradshaw et. al., 2018 is tested on a subset (73%) of the USPTO dataset instead of the full version (as explained in our paper). Therefore, we cannot directly compare with them and other methods at the same time. For this reason, we designed a separate section (the Appendix A6) to compare fairly with them. We highly recommend the reviewer to check the Appendix A6 for more details.
>
> 5) "The descriptions on the problem setup and the data are unclear. Perfectly curated as chemically correct multi-steps of bond changes are given as training set? How we can know each prediction of a bond change is correct or not?":
>
> As we presented above, the “chemically correct multi-steps of bond changes” can be easily obtained from the USPTO dataset by using the information of reactant/reagent molecules and product molecules with uniquely identified atoms. To know each prediction of bond change is correct or not, we have a ground-truth set of reaction triples. For the problem setup, we recommend the reviewer to check the Sec 2.1. In this section, we clearly presented the input, output, states, rewards, e.t.c of our problem. For example, we wrote the sentence “Given G as input, we predict a sequence of reaction triples that transforms G into a graph of product molecules G_0” to indicate that the input is the reactant/reagent molecular graph and the output is the sequence of reaction triples.
>
> [Vinyals et. al., 2016] Order Matters: Sequence to Sequence for Sets
> [Jin et. al., 2017] Predicting Organic Reaction Outcomes with Weisfeiler-Lehman Networks
> [You et. al., 2017] Graph Convolutional Policy Network for Goal-Directed Molecular Graph Generation
> [Welleck et. al., 2017] Loss Functions for Multiset Prediction
> [Bradshaw et. al., 2018] Predicting Electron Paths

---

### Meta-Review · Area_Chair1 · 2018-12-14
**After modest discussion, no clear signal for acceptance**

**Confidence:** 3
**Recommendation:** Reject

**Metareview:**

The reviewers and authors participated in modest discussion, with the authors providing direct responses to reviewer comments. However, this did not appreciably change the overall ratings of the paper (one reviewer raised their rating, while another grew more concerned), and in aggregate the reviewers do not recommend that the paper meets the bar for acceptance.